# TRIP12 structures reveal HECT E3 formation of K29 linkages and branched ubiquitin chains

Samuel A. Maiwald ●[1,2], Laura A. Schneider ●[2,4], Ronnald Vollrath[2], Joanna Liwocha[2,5], Matthew D. Maletic ●[3], Kirby N. Swatek ●[2,6], Monique P. C. Mulder ●[3] & Brenda A. Schulman ●[1,2] ✉

Regulation by ubiquitin depends on E3 ligases forging chains of specific topologies, yet the mechanisms underlying the generation of atypical linkages remain largely elusive. Here we utilize biochemistry, chemistry, and cryo-EM to define the catalytic architecture producing K29 linkages and K29/K48 branches for the human HECT E3 TRIP12. TRIP12 resembles a pincer. One pincer side comprises tandem ubiquitin-binding domains, engaging the proximal ubiquitin to direct its K29 towards the ubiquitylation active site, and selectively capturing a distal ubiquitin from a K48-linked chain. The opposite pincer side—the HECT domain—precisely juxtaposes the ubiquitins to be joined, further ensuring K29 linkage specificity. Comparison to the prior structure visualizing K48-linked chain formation by UBR5 reveals a similar mechanism shared by two human HECT enzymes: parallel features of the E3s, donor and acceptor ubiquitins configure the active site around the targeted lysine, with E3-specific domains buttressing the acceptor for linkage-specific polyubiquitylation.

E3 ligases modify proteins with specific ubiquitin (Ub) 'chains' that determine the fates of their substrates[1]. Ub chains are forged when an E3 ligase promotes transfer of the carboxy terminus of a 'donor' Ub to one of seven lysines or the amino terminus on an 'acceptor' Ub[2–10]. These chains can be homotypic, meaning that all Ubs are linked through the same site, or branched, which is when a Ub already incorporated into a chain receives additional modifications at a second site[11–13]. Linkages to different Ub sites signify distinct functions. For example, K29-linked chains are associated with proteotoxic stress responses[14–18]. K48-linked chains trigger proteasomal degradation[19]. Meanwhile, branched chains with both K29 and K48 linkages have roles in the regulation of diverse substrates in biological processes ranging from responses to oxidative, lipid, and pH stresses to targeted

protein degradation[20–25]. Despite their great biological importance[26,27], our understanding of how E3s generate K29-linked Ub chains and linkage-specific branched chains remains limited.

In humans, a major E3 ligase responsible for generating K29 linkages and branched chains, TRIP12, is central to cellular signaling and human health. TRIP12 has been associated with neurodegenerative and autism spectrum disorders[28–31]. TRIP12 regulates diverse cellular pathways—including cell division, DNA-damage responses, gene expression, differentiation, and small-molecule-induced targeted protein degradation[23,32]. Interestingly, TRIP12 not only produces K29 linkages, but also forms K29-linked branches off K48-linked chains[23]. Accordingly, TRIP12 is associated with several pathways involving other E3 ligases that specifically generate K48-linked Ub chains[23,25,27,33].

[1]Department of Chemistry, School of Natural Sciences, Technical University of Munich, Garching, Germany. [2]Department of Molecular Machines and Signaling, Max Planck Institute of Biochemistry, Martinsried, Germany. [3]Department of Cell and Chemical Biology, Leiden University Medical Center, Leiden, the Netherlands. [4]Present address: ISREC, École polytechnique fédérale de Lausanne (EPFL), Lausanne, Switzerland. [5]Present address: Lyterian Therapeutics, South San Francisco, CA, USA. [6]Present address: MRC Protein Phosphorylation and Ubiquitylation Unit, University of Dundee, Dundee, UK. ✉e-mail: schulman@biochem.mpg.de

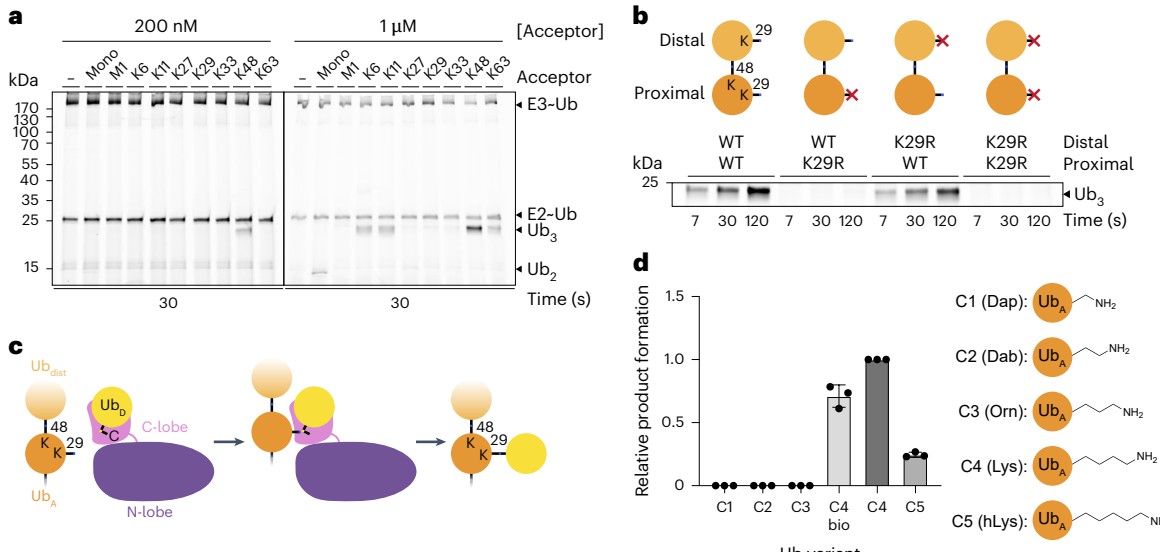

**Fig. 1 | Biochemical and chemical specificity of branched ubiquitin chain formation by TRIP12. a**, An assay testing the chain-branching specificity of TRIP12. The indicated purified di-Ubs were tested as acceptors for TRIP12-mediated production of fluorescent tri-Ub. The experiment was performed in a pulse–chase format, in which the E2-Ub intermediate formed in the pulse reaction is UBE2L3 thioester-bonded to lysineless (K0) Ub, labeled with an N-terminal fluorescent tag. The chase was initiated by adding TRIP12 together with the di-Ub with the indicated linkage. Fluorescent Ub was tracked through the cascade over time by migration in non-reducing SDS–PAGE and detected by fluorescent scan. The product (tri-Ub) is referred to as $Ub_3$ ($n = 2$ independent technical replicates). **b**, An assay showing branched ubiquitin chains produced by TRIP12 with indicated K48-linked di-Ub acceptors, with variations based on the position of K29R substitutions ($n = 2$ independent technical replicates).

**c**, Scheme of K29/K48-linked branched chain formation by TRIP12. Donor Ub ($Ub_D$)-loaded TRIP12 ubiquitylates K29 of proximal Ub in a K48 chain. The HECT domain is arranged in the ligation-specific L conformation, in which the C-lobe is left and above the horizontally viewed N-lobe. $Ub_A$, acceptor Ub; $Ub_{dist}$, distal Ub. **d**, Quantification of assays testing semi-synthetic K48-linked di-Ub substrates, varying by the indicated number of methylene groups between the α-carbon and the side chain amino group at position 29 of the proximal Ub. All variants contained synthetic Ub in the proximal position, except C4bio, which used Ub recombinantly expressed in *Escherichia coli*. Native lysine contains four methylenes. Dap, ʟ-2,3-diaminopropionic acid (one methylene); Dab, ʟ-2,4-diaminobutyric acid (two methylenes); Orn, ʟ-ornithine (three methylenes); hLys, ʟ-homolysine (five methylenes). The graph shows points from three independent technical replicates, with the means indicated by the bars. Error, s.d.

TRIP12 is a member of the founding E3 ligase family, defined by a homologous to E6AP C-terminus (HECT) catalytic domain[34,35]. E3s in this family mediate Ub transfer through multistep reactions catalyzed by distinct configurations of the bi-lobal HECT domain[36–45]. First, the HECT domain N-lobe binds an E2-Ub intermediate (here, ~ refers to a thioester bond between an enzyme catalytic Cys and Ub's C-terminus). Ub's C-terminus is then transferred from the E2 to the catalytic Cys in the HECT domain C-lobe. Earlier studies have shown that this reaction occurs with the HECT domain in an 'inverted-T conformation', in which the C-lobe faces the N-lobe-bound E2 (refs. 37,38,45). During polyubiquitylation, the HECT domain lobes rotate into an 'L conformation' (in which the C-lobe is positioned to the left and vertically when the long axis of the N-lobe is viewed horizontally) for transfer of the E3-linked donor Ub to a Lys on the acceptor Ub[9,38]. The L conformation places the E3-linked donor Ub's C-terminus in the active site, which is situated at the junction between the HECT domain N- and C-lobes, facing the acceptor[9,38,46]. However, regions of the donor and acceptor Ubs, beyond the residues that become covalently linked, have been proposed to occupy various positions relative to the HECT domain during polyubiquitylation reactions[9,38,46–49]. Thus, it remains unclear whether any subset of HECT-family E3s uses a common mechanism of polyubiquitylation. Also, the HECT E3's C-terminal amino acid has been proposed to play a key role[36,48,50]; however, this residue has not been observed in any previous structure representing a ubiquitylation reaction. Despite TRIP12's biological importance, its structure remains uncharacterized. Here, we report biochemical and structural mechanisms underlying TRIP12-catalyzed formation of K29 linkages and K29/K48-linked branched chains. Taken together with existing data, the results of our study define molecular principles of linkage-specific Ub chain formation conserved among some human HECT E3s.

## Results

### Roles of acceptor ubiquitin context and target lysine

To determine key features of TRIP12-mediated polyubiquitylation, we performed a series of biochemical pulse-chase assays that generate defined products, thereby facilitating comparisons. A fluorescently labeled donor Ub that lacks lysines and is N-terminally tagged and thus cannot be used as an acceptor (*Ub(K0)) was tracked on the basis of its migration in SDS–PAGE. *Ub(K0) was initially linked to E2 in the pulse reaction, and then transferred through TRIP12 to a specific acceptor (added with the E3 in the chase reaction). Assessment of various potential acceptors revealed that TRIP12 preferentially targets K48-linked chains, exhibiting a clear preference over di-Ubs with any other linkage or mono-Ub (Fig. 1a). TRIP12's striking selectivity for ubiquitylating K48-linked di-Ubs was retained in experiments performed with substantially higher acceptor concentrations. Under these conditions, TRIP12 did show some—but relatively less—activity toward mono-Ub and di-Ubs with K6, K11, and K63 linkages, but not others (M1, K27, K29, and K33).

The modification of both K48-linked di-Ub and mono-Ub depended on K29 of the acceptor (Extended Data Fig. 1a). Testing K48-linked di-Ubs with different combinations of K29R substitutions showed that TRIP12 preferentially modifies K29 in the proximal Ub (Fig. 1b). Altogether, the data suggest: (1) the distal Ub in a K48-linked di-Ub chain contributes to acceptor binding; (2) specificity for targeting K29 is intrinsic to TRIP12 and does not require a K48-linked Ub chain acceptor, but is restricted by some linkages; and (3) a K48-linked di-Ub is constrained with the proximal Ub placed for ubiquitylation on its K29 (Fig. 1c).

We considered that there could be tight geometric constraints affecting TRIP12's production of K29-linked Ub chains. Indeed, prior studies of some E2s and E3s showed that their ability to synthesize

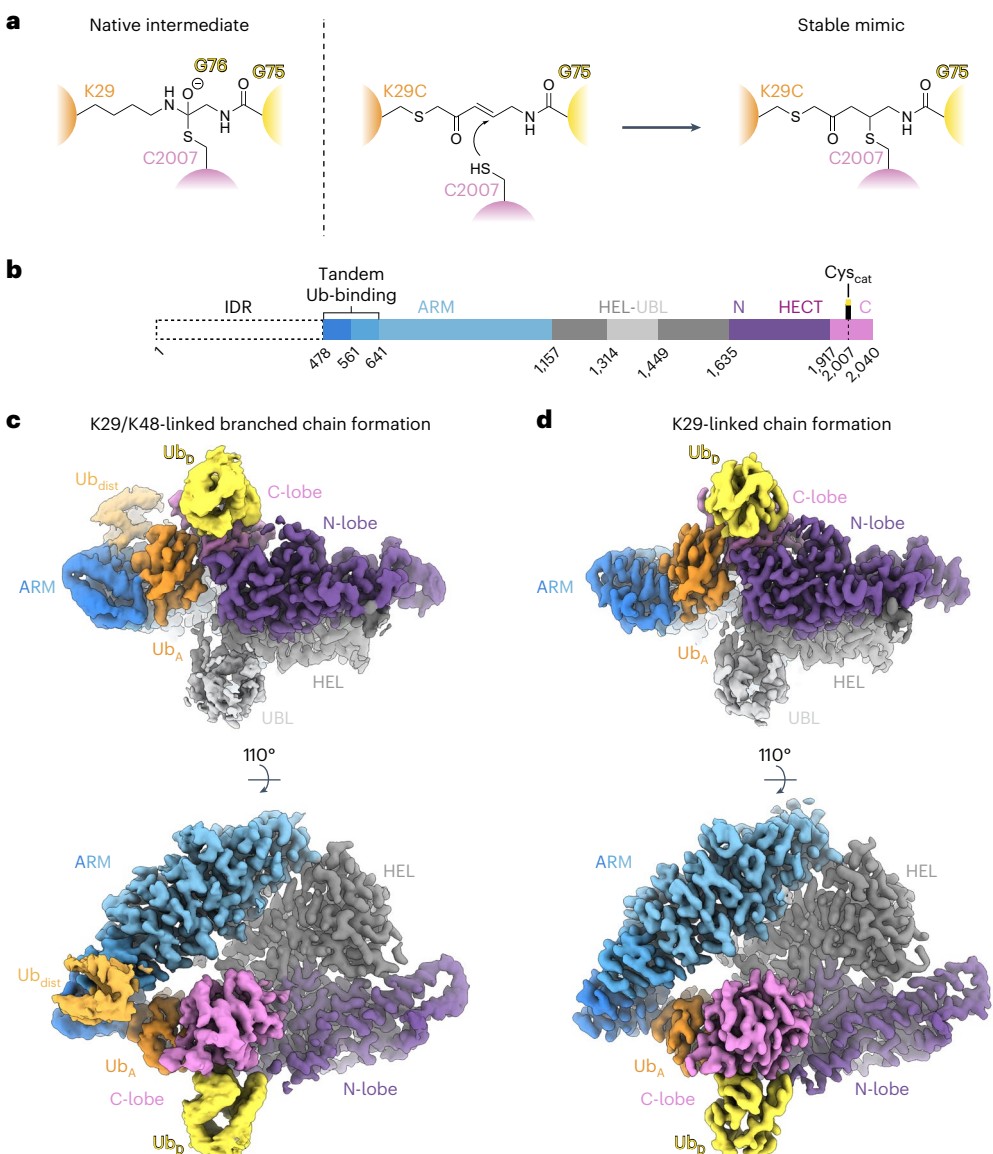

**Fig. 2 | Cryo-EM structures representing TRIP12-catalyzed polyubiquitylation.**
**a**, Chemical structures of native intermediate (left) and the stable mimic (right) involved in generating K29 linkages. **b**, Schematic of TRIP12 domains. IDR, intrinsically disordered region; ARM, armadillo repeats with tandem Ub-binding regions; HEL-UBL, helical scaffold and Ub-like domain; HECT, catalytic domain with N- and C-lobes. **c**, Cryo-EM map (DeepEMhancer-sharpened) representative of TRIP12$^{\Delta N}$ forming a K29/K48 branched ubiquitin chain. TRIP12 domains are colored according to scheme in **b**, ubiquitins according to scheme in **a**. **d**, Similar to **c**, but for complex representing the formation of a K29-linked ubiquitin chain.

K48- and K63-linked chains is exquisitely sensitive to the number of methylene groups between the α-carbon and amino group that is the site of modification[51,52]. To determine whether TRIP12 activity is also sensitive to such acceptor geometry, we created a series of semi-synthetic K48-linked di-Ub substrates. To assess the modification of different lysine analogs, we placed them at position 29 of the proximal Ub, with the distal Ub harboring a K29R substitution (Extended Data Fig. 1b). We compared the activity of the semi-synthetic and recombinant acceptors with L-lysine (four methylenes) and variants with shorter or longer side chains: one methylene (L-2,3-diaminopropionic acid), two methylenes (L-2,4-diaminobutyric acid), three methylenes (L-ornithine), or five methylenes (L-homolysine). Formation of branched chains was undetectable for acceptor side chains shorter than lysine (tetramethylene linker) and was impaired with the longer side chain (Fig. 1d). Thus, the data suggest that K29/K48-branched Ub chain formation depends on a specialized geometric arrangement in which the epsilon amino group of the acceptor lysine is positioned precisely relative to the E3-Ub active site.

## Visualizing TRIP12 forging K29- and K29/K48-linked chains

To define the underlying structural determinants, we reasoned that the donor and acceptor Ubs are only transiently juxtaposed during branched chain formation. Thus, we adapted our strategy to capture stable mimics representing transition states during ubiquitylation[9,10,53]. In brief, TRIP12's active site Cys2007 was stably linked to a chemical warhead installed between the donor Ub's C-terminus and K29C of the proximal Ub in a K48-linked di-Ub chain. Importantly, our chemical biology tool maintains the native number of bonds between the TRIP12 catalytic Cys, the donor Ub's penultimate residue G75, and the α-carbon of the acceptor site (Fig. 2a). By subjecting the complex to cryo-electron microscopy (cryo-EM), we obtained a map revealing the overall assembly (Table 1 and Extended Data Fig. 2). The structure resembles a pincer clamped around the acceptor Ub. The two sides are connected by a central HEL-UBL domain, which is largely helical but also has a Ub-like fold insertion. The N-terminal Armadillo-repeat (ARM) domain serves as one side of the pincer. The opposite side consists of

**Table 1 | Data collection, refinement and validation statistics**

| | Full-length TRIP12, K29/K48 branched chain formation complex (EMDB-51428) | TRIP12$^{\Delta N}$, K29/K48 branched chain formation complex (screening dataset) | TRIP12$^{\Delta N}$, K29/K48 branched chain formation complex (EMDB-51429) (PDB 9GKM) | TRIP12$^{\Delta N}$, K29 chain formation complex (screening dataset) | TRIP12$^{\Delta N}$, K29 chain formation complex (EMDB-51430) (PDB 9GKN) |
|---|---|---|---|---|---|
| **Data collection and processing** | | | | | |
| Magnification | ×105,000 | ×22,000 | ×105,000 | ×22,000 | ×105,000 |
| Voltage (kV) | 300 | 200 | 300 | 200 | 300 |
| Electron exposure (e⁻ per Å$^2$) | 76.5 | 60.0 | 64.8 | 60.0 | 66.8 |
| Defocus range (μm) | −0.6 to −2.2 | −0.8 to −2.6 | −0.6 to −2.2 | −0.8 to −2.6 | −0.6 to −2.2 |
| Pixel size (Å) | 0.8512 | 1.841 | 0.8512 | 1.841 | 0.8512 |
| Symmetry imposed | $C_1$ | $C_1$ | $C_1$ | $C_1$ | $C_1$ |
| Initial particle images (no.) | 6,505,263 | 1,319,552 | 939,881 | 1,871,755 | 5,689,483 |
| Final particle images (no.) | 891,840 | 573,040 | 122,281 | 358,901 | 427,215 |
| Map resolution (Å) | 3.2 | 4.3 | 3.7 | 6.9 | 3.4 |
| Fourier shell correlation threshold | 0.143 | 0.143 | 0.143 | 0.143 | 0.143 |
| Map resolution range (Å) | 1.8–49.1 | 4.1–12.3 | 3.1–14.4 | 5.1–61.0 | 1.8–15.1 |
| **Refinement** | | | | | |
| Initial model used (PDB code) | | | AlphaFold 2 multimer | | AlphaFold 2 multimer |
| Model resolution (Å) | | | 3.7 | | 3.4 |
| Fourier shell correlation threshold | | | 0.143 | | 0.143 |
| Model resolution range (Å) | | | 3.1–14.4 | | 1.8–15.1 |
| Map sharpening B factor (Å$^2$) | | | – | | – |
| Model-map correlation (box) | | | 0.74 | | 0.76 |
| Model composition | | | | | |
| Non-hydrogen atoms | | | 11,231 | | 10,763 |
| Protein residues | | | 1446 | | 1384 |
| Ligands | | | 1 (SY8) | | 1 (SY8) |
| B factors (Å$^2$) | | | | | |
| Protein | | | 77.5 | | 74.2 |
| Ligand | | | 81.3 | | 62 |
| Root mean square deviations | | | | | |
| Bond lengths (Å) | | | 0.002 | | 0.003 |
| Bond angles (°) | | | 0.497 | | 0.548 |
| Validation | | | | | |
| MolProbity score | | | 1.71 | | 1.63 |
| Clashscore | | | 6.29 | | 4.85 |
| Poor rotamers (%) | | | 0 | | 0 |
| Ramachandran plot | | | | | |
| Favored (%) | | | 94.7 | | 94.55 |
| Allowed (%) | | | 5.3 | | 5.45 |
| Disallowed (%) | | | 0 | | 0 |

the HECT domain N- and C-lobes in the L configuration (see Fig. 2b for domain annotation). Although anisotropy due to preferred particle orientation limited local resolution around the active site (Extended Data Fig. 2), the map revealed that the donor and acceptor Ubs were uniquely splayed across the catalytic HECT domain, establishing K29 linkage specificity and a preference for branching off K48-linked chains (Supplementary Video 1).

Higher-resolution insights—including details of the active site—were obtained with a truncated version of TRIP12, termed TRIP12$^{\Delta N}$.

TRIP12$^{\Delta N}$ lacks the intrinsically-disordered N-terminal region (residues 1–477) that was not visible in the map of the full-length complex. TRIP12$^{\Delta N}$ maintains K29 linkage specificity and a preference for the K48-linked di-Ub substrate (Extended Data Fig. 1c). As the distal Ub in the K48-linked chain was observed only at relatively lower contour than the proximal acceptor Ub in our initial map (Extended Data Fig. 2), we prepared two samples for cryo-EM. One sample represents the transition state for TRIP12$^{\Delta N}$ generating a branched K29/K48-linked chain. The other represents Ub linkage to K29 on a mono-Ub acceptor.

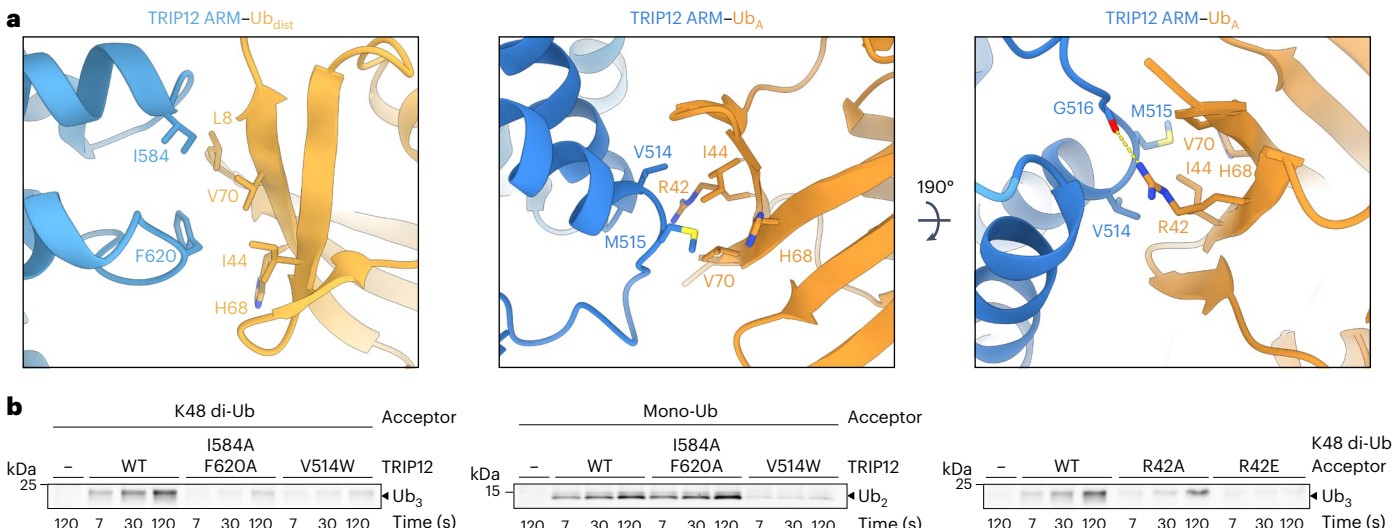

**Fig. 3 | TRIP12 specificity for K29 branching off of K48 chain acceptors.**
**a**, Close-ups showing TRIP12 ARM domain interfaces with K48-linked distal (left) and acceptor (center and right) Ubs. **b**, Pulse–chase assays testing the effects of TRIP12 ARM domain substitutions on formation of a K29/K48-linked tri-ubiquitin branched chain (left, using K48-linked di-Ub as acceptor) or a K29-linked di-ubiquitin chain (center, using mono-Ub acceptor), and the effects of R42 substitutions in the proximal Ub of a di-Ub acceptor on formation of a branched tri-Ub (right, *n* = 2 independent technical replicates).

These yielded superior cryo-EM maps with overall resolutions of 3.7 Å and 3.3 Å (Table 1, Fig. 2c,d and Extended Data Figs. 3–5). The maps obtained with TRIP12$^{\Delta N}$ superimpose with each other and the map of the full-length complex (Supplementary Video 1). The data facilitated the visualization of the juxtaposition of the acceptor and donor Ubs and key HECT E3 catalytic elements (Supplementary Video 2), and enabled atomic model building (Extended Data Fig. 6). The results are collectively described below.

**Tandem ubiquitin binding domains establish chain branching**
TRIP12's preference for a K48-linked Ub chain substrate was revealed by the structures representing branched chain formation. The I44-centered hydrophobic patches of both Ubs in the K48-linked substrate bind adjacent surfaces on TRIP12's ARM repeats (Fig. 3a and Extended Data Fig. 7a). These interactions resemble a K48-linked di-Ub complex binding to the yeast TRIP12 ortholog Ufd4 (ref. 49) (Extended Data Fig. 7b). The arrangement of TRIP12's tandem Ub binding sites is exclusively compatible with a K48-linked chain, not other linkages (Extended Data Fig. 7c). The tandem Ub-binding domain organization is observed even in the complex featuring mono-Ub modification (Fig. 2d and Supplementary Video 1). The positioning of the adjacent, vacant site, ready to capture the distal Ub in a K48-linked chain despite its absence, accounts for the striking preference for branching off chains linked through K48, rather than other lysines.

The structure also explains why the proximal Ub is the acceptor: it faces the ubiquitylation active site (Fig. 2c). Indeed, substitutions in either Ub-binding site on the ARM domain impaired TRIP12-mediated Ub transfer to K48-linked di-Ub (Fig. 3b). Only substitutions in the proximal Ub-binding site substantially impacted TRIP12 activity toward a mono-Ub substrate, in accordance with our biochemical data showing that the proximal Ub is the acceptor. The mutational data also agree with our structure showing this ARM region binds the acceptor mono-Ub for TRIP12 generation of K29-linked di-Ub (Figs. 2d and 3a). Additionally, substitution of the acceptor Ub's R42, which contacts the proximal Ub-binding site in the structure, also impaired modification by TRIP12 (Fig. 3a,b and Extended Data Fig. 8).

The structure suggests TRIP12 could engage—and thus target—di-Ubs along a K48-linked chain. We tested this using pulse-chase assays examining TRIP12 modification of longer chains. K29-linked chains (largely tetra-Ub, but with some tri-Ub) and K48-linked chains

(primarily tetra-Ub and a minor amount of penta-Ub) were biochemically prepared for testing as TRIP12 substrates. Ubiquitylation was initiated with E2-*Ub(K0) either equimolar with or in fourfold excess of acceptors. Mono-Ub was converted to di-Ub, but modification of the K29-linked chain was not readily detected. Strikingly, however, TRIP12 robustly added multiple mono-Ubs to the K48-linked chains (Extended Data Fig. 9a). We also compared TRIP12 activity with E2-Ub complexes harboring either untagged K29R Ub or wild-type (WT) Ub as the donor, as the latter in principle permits its subsequent use as an acceptor during polyubiquitylation (Extended Data Fig. 9b). For these assays, immunoblotting with anti-Ub antibodies detects Ubs from both the donor and the acceptor moieties. The products generated with the K29R donor Ub largely resembled those in assays with *Ub(K0), while additional bands and variations in their relative intensities were observed in assays with WT donor Ub. Thus, to determine the nature of these polyubiquitin chains, we tested their cleavage by linkage-specific deubiquitylating (DUB) enzymes (Extended Data Fig. 9c). The main products were confirmed as mono-Ubs linked to K29 of acceptors in the K48-linked chain because treatment with OTUB1* (a K48-linkage-specific DUB[16]) collapsed the majority of products to di-Ub. Susceptibility of the OTUB1*-generated di-Ub products to deconjugation by TRABID (a K29-linkage-specific DUB[16,17]) confirmed the K29 linkages. Taken together, the data support the conclusion that TRIP12 preferentially branches K29 linkages from K48-linked di-Ubs, including within a polyUb chain.

**Elaborate interactions arrange TRIP12 catalytic architecture**
On the opposite side of the acceptor, numerous protein-protein interactions converge to juxtapose the acceptor Ub's K29 and donor Ub's C-terminus in the active site. The acceptor Ub's residue 29 (here a chemically modified Cys, but normally a Lys) is situated in a fiveway junction together with the donor Ub and the C- and N-lobes and extreme C-terminal element of TRIP12's HECT domain (Fig. 4a). In this arrangement, the acceptor Ub's N-terminus and K33 abut the HECT-Ub intermediate, and K27 is buried in the acceptor Ub itself (Extended Data Fig. 7d). Thus, the structure explains why pre-existing linkages through these residues (and obviously K29) impede usage as an acceptor for TRIP12.

The acceptor Ub's helix, which contains K29, aligns across the surface created by the HECT domain's N- and C-lobes in the L conformation. As such, the acceptor Ub's acidic face (the region spanning

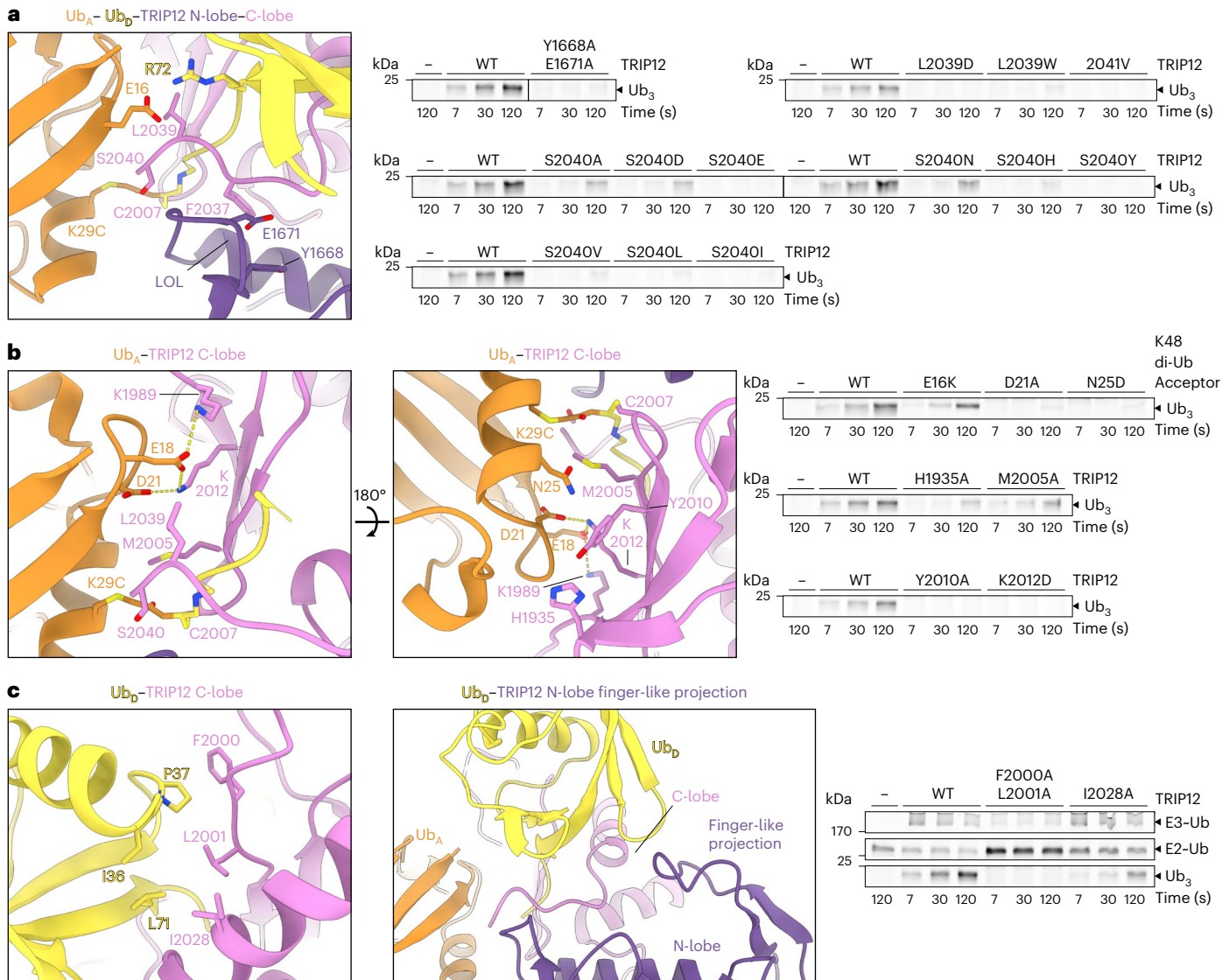

**Fig. 4 | Elaborate interactions between the HECT domain, acceptor and donor ubiquitins establish the active site. a**, Left: close-up showing the extensive contacts between the TRIP12 N-lobe's ligation-organizing-loop (LOL, purple) and extreme C-terminus (residues F2037–S2040, violet) with each other, the C-lobe, the donor Ub (yellow), and the acceptor Ub (orange). Right: assays testing the effects of substitutions in the TRIP12 LOL and C-terminus (including adding a residue, 2041V) on the formation of a K29/K48-linked tri-ubiquitin branched chain. Reactions were performed in pulse–chase format using full-length TRIP12, and K48-linked di-Ub as acceptor. **b**, Left: Close-ups showing the interactions between TRIP12's C-lobe and acceptor Ub (orange). Right: assays testing the effects of the indicated substitutions in the proximal acceptor Ub in a K48-linked di-Ub target, or in TRIP12, on generating a branched tri-Ub chain. **c**, Left and center: close-ups showing interactions between TRIP12's C- and N-lobes and donor Ub (yellow). Right: assay testing the effects of the indicated substitutions in TRIP12 on generating a branched tri-Ub chain. **a**–**c**, All gels are representative of *n* = 2 independent technical replicates.

from E16 to D21, preceding the helix) nestles in the complementary concave surface formed by the HECT domain C-lobe β-sheet, TRIP12's two C-terminal residues, and the donor Ub (Fig. 4a,b). Here, TRIP12's penultimate L2039 serves as a hydrophobic glue between the acceptor and donor Ubs. TRIP12's C-terminal Ser also fits snugly in the interface between the acceptor Ub and TRIP12's N- and C-lobes, aligning the donor Ub's covalent bond with TRIP12's catalytic Cys. The resolution of our map precludes distinguishing the positions of the C-terminal Ser side chain and charged carboxylate. Nonetheless, this visualization of a HECT E3 C-terminal residue wedged in the active site of a ubiquitylation complex is consistent with its proposed structural and catalytic roles during Ub chain formation[50].

The donor Ub's C-terminal tail forms a strand between the C-lobe β-sheet and the E3's extreme C-terminus that is itself stabilized by a conserved N-lobe element referred to as the 'ligation organizing loop'[9,38]. The donor Ub is further positioned by non-covalent interactions with TRIP12's C-lobe and with a finger-like projection from the N-lobe (Fig. 4c).

Of note, substitutions at key interfaces impaired TRIP12-mediated branched Ub chain formation (Fig. 4, right side panels). The importance of the C-terminal Ser tightly fitting in the catalytic interface was highlighted by the deleterious effects following its replacement with corresponding residues (A, D, E, H, I, L, N, V, and Y) found in other HECT E3s. Substitutions at the C-lobe interface with the donor Ub were defective, but this region is also involved in forming a stable HECT E3-Ub intermediate (Fig. 4c, right panel); we were unable to express TRIP12 mutants lacking the finger-like projection. Notably, reactions with acceptor mono-Ub (performed at a higher concentration) showed that the substitutions in the catalytic interfaces impaired K29-dependent di-Ub formation (Extended Data Fig. 8), as well as chain branching.

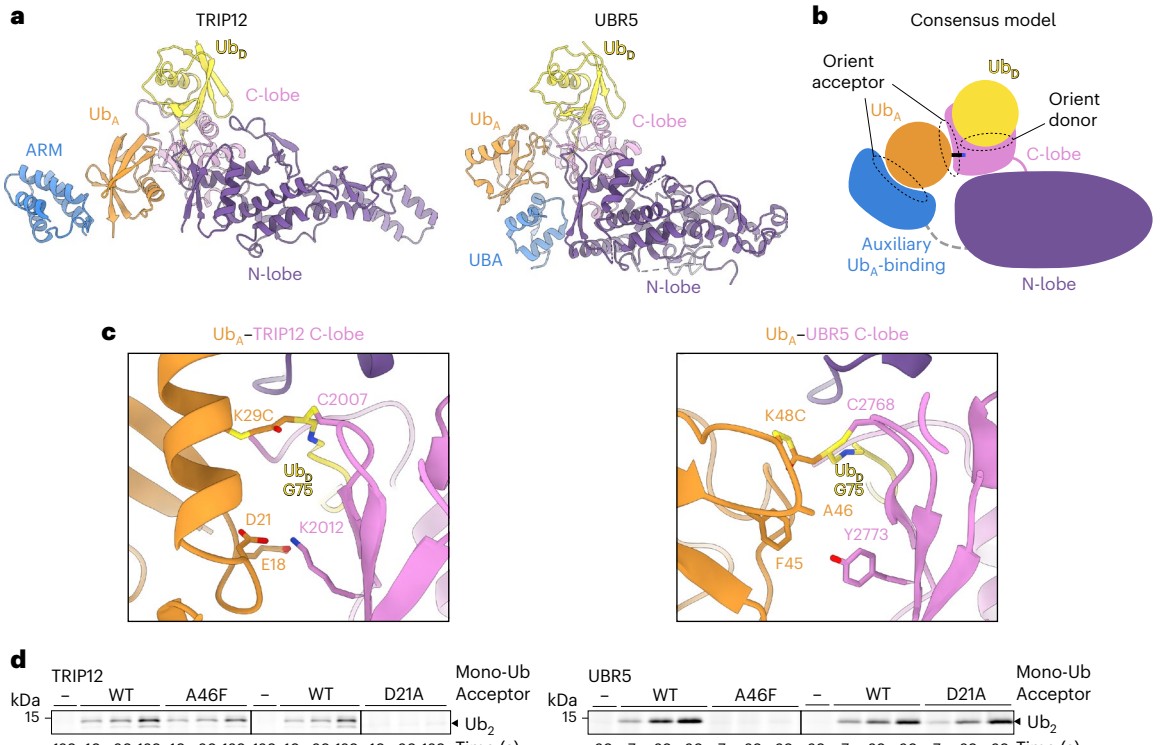

**Fig. 5 | Conserved catalytic architecture for linkage-specific polyubiquitylation by human HECT E3 ligases. a**, Structures of the TRIP12 (left) and UBR5 (right, PDB ID: 8C07) catalytic assemblies forming K29 and K48 linkages. The structures were aligned over their HECT domain C-lobes. **b**, Cartoon showing consensus architecture for TRIP12 and UBR5 HECT E3 alignment of donor and acceptor Ubs establishing linkage-specific chain synthesis. **c**, Close-ups over the $Ub_A$ interface with TRIP12-$Ub_D$ (left) and UBR5-$Ub_D$ (right), aligned over the C-lobes, similar to Fig. 4b. **d**, Assays testing the effects of substitutions at residues in the acceptor Ub that form non-covalent interactions specifically with the HECT E3 C-lobe from TRIP12 or UBR5. D21 is central to the acceptor

Ub interface with the TRIP12 C-lobe, but is surface-exposed in complex with UBR5. A46 is central to the acceptor Ub interface with the UBR5 C-lobe, but is surfaced-exposed in complex with TRIP12. The experiment was performed in pulse–chase format, in which the E2 of the E2-Ub intermediate is thioester-bonded to fluorescently labeled, lysineless (K0) Ub, formed in the pulse reaction. E2 was UBE2L3 in reaction with TRIP12, and UBE2D2 in reaction with UBR5, on the basis of the catalytic preferences for these E3s. The chase was initiated by adding indicated E3 together with a mono-Ub acceptor. The di-Ub product is labeled $Ub_2$ ($n$ = 2 (for TRIP12) and $n$ = 3 (for UBR5)) independent technical replicates).

Thus, TRIP12's exquisite linkage specificity can be explained by the configuration of the HECT domain and donor and acceptor Ubs, together with the proximal Ub-binding site on the ARM domain.

Synergistic roles for key TRIP12 regions—and the donor and acceptor Ubs—in organizing the catalytic configuration also emerged from 3D variability analysis[54] of the complex representing branched chain formation (Supplementary Video 2). Two distinct sub-assemblies were maintained throughout the maps: (1) a major portion of TRIP12, encompassing the ARM, HEL-UBL, and HECT N-lobe domains; and (2) the C-lobe–acceptor Ub assembly. Advancing through successive frames revealed a coordinated increase in visibility for both the donor Ub and TRIP12's extreme C-terminus, concomitant with progression of the C-lobe and acceptor Ub toward the N-lobe. As such, the donor Ub and TRIP12 C-terminus are most clearly visible in the configuration in which the key TRIP12 N-lobe, C-lobe, C-terminal tail, and donor and acceptor Ub elements required for polyubiquitylation are intertwined.

### A polyubiquitylation mechanism shared by some human HECT E3s

To ascertain common and unique mechanisms of polyubiquitylation across HECT E3s, we compared the TRIP12 assembly with that of UBR5. UBR5 forges K48-linked chains and is presently the only other HECT E3 for which a structure representing polyubiquitylation has been published[9]. A conserved configuration for site-specific Ub transfer from a HECT E3 to a precisely placed acceptor emerged from superimposing the structures over their C-lobes (Fig. 5a,b). This consensus assembly

is defined by: (1) an auxiliary E3 domain recruiting the acceptor Ub on a surface distal from its target Lys, specifically, the acceptor Ub's I44 patch binds ARM repeats in TRIP12 and the UBA domain in UBR5; (2) the C-lobe orienting the donor Ub through a conserved interface with this Ub's I36 hydrophobic patch; (3) the HECT domain C-lobe adopting the L configuration relative to the N-lobe; (4) the resultant placement of the donor Ub's C-terminus at the nexus between the E3's catalytic Cys in the C-lobe, extreme C-terminal element, and the ligation organizing loop from the N-lobe; (5) the C-lobe β-sheet, together with the donor Ub (R72 for both TRIP12 and UBR5), forming an extensive network of interactions with the acceptor that places the target Lys into the active site; and (6) when the HECT E3's penultimate residue is hydrophobic as in TRIP12 and UBR5, this serving as a glue bridging the donor and acceptor Ubs.

Furthermore, our data visualize how linkage specificity is conferred in part by distinct interactions between the C-lobe β-sheet and the acceptor Ub's surface adjacent to its target Lys. A positively charged residue (K2012) in TRIP12 contacts the complementary acidic E18 and D21 residues on the acceptor Ub (Figs. 4b and 5c). The corresponding UBR5 residue (Y2773) is hydrophobic, and contacts F45 and A46 of its acceptor Ub (Fig. 5c). Accordingly, the D21A substitution impairs Ub's modification by TRIP12 but not UBR5; the A46F substitution impairs Ub's modification by UBR5 but not TRIP12 (Fig. 5d).

### Discussion

Overall, our study extends knowledge of the ubiquitin system, by (1) establishing the mechanism of TRIP12, an important human Ub

ligase; (2) illuminating the formation of branched chains at high resolution; and (3) defining mechanisms underlying linkage-specific polyubiquitylation shared by the two characterized members of a major human E3 family.

Our biochemical data define key elements of TRIP12-catalyzed K29/K48-linked branched chain formation: TRIP12 intrinsically generates K29 linkages, and most efficiently modifies the proximal Ub in K48-linked ubiquitin chains. Importantly, this activity depends on the length of the lysine side chain at residue 29, as evidenced by reactions in which the acceptors were semi-synthetic K48-linked di-Ubs with unnatural amino acids substituted at K29 (Fig. 1). We also report an activity-based chemical tool for branched ubiquitin chain formation that maintains the critical native distance of an acceptor lysine (Fig. 2a).

This tool enabled the visualization of TRIP12 as if in the act of synthesizing a branched Ub chain. Elements from the HECT domain N-lobe, C-lobe and also TRIP12's C-terminal region intertwine with the donor and acceptor Ubs in a manner that exclusively places the acceptor K29 in the active site (Figs. 2–4). Structure-based mutants confirmed the key elements in HECT E3-catalyzed generation of linkages to Ub's K29 (Extended Data Fig. 8). Meanwhile, the pincer-like structure of TRIP12 both braces the acceptor from the opposite side and selectively engages an adjacent Ub in a K48-linked chain (Figs. 2 and 3). This arrangement determines TRIP12's preferred activity.

Interestingly, the structural data suggest a consensus mechanism for how at least some HECT E3s forge linkage-specific Ub chains. Comparing the structure of TRIP12 producing a K29/K48-linked branched chain with the only other high-resolution structure showing polyubiquitylation by a HECT E3 (human UBR5 generating a K48-linked chain) revealed commonalities in the catalytic configuration (Fig. 5). The C-lobes of TRIP12 and UBR5 form homologous interactions with the donor Ub. This arrangement was observed in numerous crystal structures of complexes between HECT domains and Ub[9,37,38,46–48], potentially reflecting a typical catalytic arrangement (Extended Data Fig. 10a). Furthermore, prior mutational analyses of the acceptor Ub during K29- or K48-linked chain formation by the dual-specificity HECT E3 KIAA10 (UBE3C)[55] implicated the surfaces corresponding to those recognized by the C-lobes of TRIP12 and UBR5, respectively. The parallel C-lobe β-sheet region is also crucial for generation of K63-linked Ub chains by NEDD4-family E3s[38,56]. However, moderate-resolution cryo-EM maps for the budding yeast Ufd4 showed its C-lobe interacting with human donor and acceptor Ubs in different relative arrangements[49], with fewer contacts than observed with TRIP12 or UBR5 (Extended Data Fig. 10). These disparities could reflect distinctions between human and yeast proteins, and/or the complex containing subunits from different species[49]. Alternatively, the Ufd4 structure could represent a state along the dynamic conformational progression of the reaction, perhaps captured through use of a distinct chemical tool. Nonetheless, taken together, the structures of TRIP12 and other HECT complexes illuminate how distinct sequences displayed on a common E3 fold can determine the linkage specificity of polyubiquitylation, and how these are elaborated in TRIP12 to synthesize linkage-specific branched chains.

Finally, our study has implications for the biological activity of TRIP12 as an 'E4 enzyme', preferentially modifying pre-ubiquitylated substrates[57]. Our structural and biochemical data provide mechanistic underpinnings for biological studies associating TRIP12 with ubiquitylation cascades that generate K48-linked chains. TRIP12 contributes to targeted protein degradation pathways that also rely on the cullin-RING ligase CRL2[VHL], which mediates ubiquitylation by the K48-linkage specific E2 UBE2R2 (refs. 10,23,25,58). TRIP12 also contributes to the oxidative stress response orchestrated by CRL3[KEAP1], which might use a similar cullin–RING ligase mechanism to generate K48-linked Ub chains[27]. Furthermore, TRIP12 partners with the K48-linked chain-forming HECT E3 UBR5 to regulate double-stranded DNA-break repair[59]. Notably, both the K48-linked chain-forming enzymes UBE2R2 and UBR5

show strong, reciprocal codependency with TRIP12 in DEPMAP[9,10,60]. On the basis of the functional—and now mechanistic—precedent established by TRIP12 and its partner ubiquitylation pathways, we anticipate that many exciting future studies will show how distinct chain linkages are combined. It seems likely that sequential or concerted actions of multiple ubiquitylation enzymes will prove to be a common mechanism tuning biological regulatory pathways.

## Online content

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

## Methods

### Cloning, protein expression and purification

All constructs in this study were generated using standard molecular biology techniques. Mutant constructs were generated by QuikChange Site-Directed Mutagenesis, using the oligonucleotides listed in Supplementary Table 1. Construct sequences were verified through DNA sequencing (Microsynth Seqlab). Constructs for insect cell expression were cloned into pLIB vectors[61] to allow generation of bacmids in DH10EmBacY[62]. Subsequently, baculoviruses were generated in Sf9 insect cells (Thermo Fisher Scientific cat. no. B85502) and were further used to infect Hi5 insect cells (BTI-TN-5B1-4; Thermo Fisher Scientific, cat. no. 11496016) for expression.

Bacterial expression of Ub, Ub variants, E2 UBE2L3, and DUBs OTUB1* and TRABID was performed by transforming corresponding plasmids into *E. coli* BL21(DE3) Rosetta cells. Single colonies were picked to inoculate precultures in LB medium supplemented with appropriate antibiotics, which were further diluted 1:100 into TB medium containing antibiotics. Cells were grown at 37 °C and 180 rpm to an optical density of 0.8; then, the temperature was lowered to 16 °C, and expression was induced by addition of 0.5 mM isopropyl β-ᴅ-thiogalactopyranoside (IPTG). After 18 h, cells were collected by centrifugation at 7,278*g* and 4 °C for 15 min.

Pellets were resuspended in ice-cold lysis buffer (50 mM Tris-HCl, pH 8, 200 mM NaCl, 5 mM DTT (or β-mercaptoethanol for histidine-tagged constructs) and 2.5 mM PMSF, and additionally 10 µg ml⁻¹ leupeptin, 20 µg ml⁻¹ aprotinin and 10 µg ml⁻¹ DNAse I for insect cells). Cells were lysed through sonication followed by centrifugation at 50,000*g* for 30 min to separate cell debris.

C-terminally hexahistidine-tagged Ub constructs (Ub–6×His) were purified as described previously[9]. In brief, lysates were subjected to Ni-NTA affinity chromatography, followed by size-exclusion chromatography in 25 mM HEPES, pH 7.5, and 150 mM NaCl.

GST–human rhinovirus 3C protease cleavage site (3C)–Ub variants (GST–3C–Cys–Ub^K6R K11R K27R K29R K33R K48R K63R for fluorescent labeling, and GST–3C–Ub^K29C for the K29-linked chain formation probe) were pulled down through GSH affinity chromatography, followed by cleavage with 3C protease, leaving a residual GP dipeptide at the N-terminus (or GPC for Cys–Ub^K6R K11R K27R K29R K33R K48R K63R (Ub K0)). Cleaved Ubs were further purified by size-exclusion chromatography in 25 mM HEPES, pH 7.5, 150 mM NaCl, and 1 mM DTT. The former variant, when fluorescently labeled as described below, is referred to as *Ub(K0) in the main text and is used in pulse–chase ubiquitylation assays.

Untagged Ub and variants were purified through acetic acid precipitation. Lysates were acidified by the gradual addition of acetic acid; they were then stirred at room temperature until the pH reached ~4.5 and centrifuged at 20,000 rpm for 30 min. The supernatant was dialyzed against 25 mM sodium acetate, pH 4.5, at 4 °C overnight, and was then centrifuged again at 20,000 rpm for 30 min to clear any further aggregates. The supernatant was subjected to cation exchange chromatography followed by size-exclusion chromatography in 25 mM HEPES, pH 7.5, and 150 mM NaCl.

The E2 UBE2L3 was expressed as a N-terminal GST fusion with a 3C protease cleavage site and purified by GSH affinity chromatography, followed by on-bead cleavage with 3C protease overnight. The cleaved protein eluate was then subjected to anion exchange chromatography, during which UBE2L3 was collected in the flowthrough, and size exclusion chromatography in 20 mM Tris, pH 7.5, 150 mM NaCl, and 1 mM DTT.

OTUB1* was purified as described previously[16], using Ni-NTA affinity chromatography followed by size exclusion chromatography in 25 mM HEPES, pH 7.5, 150 mM NaCl, and 2 mM DTT.

The TRABID catalytic domain (residues 245–697) was expressed as an N-terminal GST fusion with an intervening tobacco etch virus (TEV) protease cleavage site and purified through GSH affinity chromatography. Bound protein was eluted and further purified by anion exchange chromatography, followed by size-exclusion chromatography in 25 mM HEPES, pH 7.5, 150 mM NaCl, and 2 mM DTT.

The E1 UBA1 was expressed in insect cells as an N-terminal GST fusion with a TEV protease cleavage site and isolated by GSH affinity chromatography, followed by TEV protease cleavage overnight. Cleaved protein was further purified by anion exchange chromatography and subsequent size-exclusion chromatography in 25 mM HEPES pH 7.5, 150 mM NaCl, and 1 mM DTT.

TRIP12 constructs were expressed in insect cells as N-terminal GST fusions with a TEV protease cleavage site. For full-length TRIP12 (and corresponding point mutants), additional NaCl was added to resuspended pellets to a final concentration of 1 M before sonication. Protein was purified using GSH affinity chromatography and cleaved on beads by incubation with TEV protease overnight. Cleaved protein was washed off the beads and subjected to anion exchange chromatography, followed by size-exclusion chromatography in 25 mM HEPES, pH 7.5, 150 mM NaCl, and 1 mM TCEP. The TRIP12^478–2040 variant (termed TRIP12^ΔN, lacking the N-terminal intrinsically disordered region) was purified in a similar manner, but NaCl was not added before lysis and cation exchange chromatography was used.

WT UBR5 was prepared as described previously[9]. In brief, baculoviruses with a TwinStrep-GFP-UBR5 construct, generated in Sf9 cells, were used for expression in HEK293S cells (CRL-3022, ATCC). UBR5 was purified through Strep affinity chromatography, overnight cleavage using 3C protease, and subsequent size-exclusion chromatography in 25 mM HEPES, pH 7.5, 150 mM NaCl, and 1 mM TCEP.

Synthetic Ubs containing lysine or lysine analogs with different side chain lengths (one methylene (Dap), ʟ-2,3-diaminopropionic acid; two methylenes (Dab), ʟ-2,4-diaminobutyric acid; three methylenes (Orn), ʟ-ornithine; or five methylenes (hLys), ʟ-homolysine) in position 29, and an additional aspartate at the C-terminus for use in di-Ub generation (see below), were synthesized through solid-phase peptide synthesis, as described in Supplementary Note 1.

### Generation of Ub chains with different linkage types

K27-, K29-, and K33-linked di-Ubs were synthesized as reported previously[9]. M1-linked di-Ub was produced recombinantly as a linear fusion with an N-terminal GST-tag in *E. coli* and purified as described previously[9].

K6-, K11-, K48-, and K63-linked Ub chains were prepared as follows, using enzymatic assembly with recombinantly produced tagless Ub.

K6-linked chains were generated by incubating 2.5 mM Ub with 0.1 µM E1, 0.6 µM UBE2L3, 10 µM NleL in 40 mM Tris-HCl, pH 8.8, 10 mM MgCl₂, 1 mM DTT, and 10 mM ATP for 3 h at 37 °C. The reaction was quenched with 10 mM DTT, and by-product K48-linked Ub chains were removed by subsequent incubation with 2 µM OTUB1 for 3 h at 37 °C.

K11-linked Ub chains were generated by incubating 0.5 mM Ub with 0.25 µM E1 and 5 µM Ube2S-UBA-IsoT[63] in the presence of 10 mM ATP for 2 h at 37 °C.

To generate native K48-linked Ub chains, 2.5 mM Ub was incubated with 1 µM E1 and 25 µM UBE2R1 in the presence of 10 mM ATP for 3 h at 37 °C. The reaction was quenched by adding 10 mM DTT and 1 µM associated molecule with the SH3 domain of STAM (AMSH).

K63-linked Ub chains were generated by incubating 1 mM Ub with 0.5 µM E1, 8 µM Ube2N, and 8 µM Ube2V1 in 40 mM Tris-HCl, pH 8.5, 10 mM MgCl₂, 0.5 mM DTT, and 10 mM ATP at 37 °C for 30 min, and were then quenched by the addition of 10 mM DTT.

Different chain lengths of the various chain types were consecutively separated using several rounds of cation exchange chromatography, followed by size-exclusion chromatography in 25 mM HEPES pH 7.5, of which only di-Ubs were used in this study.

### Generation of mutant K48-linked di-Ubs

Specific mutant K48-linked di-Ubs were produced by incubating 275 µM donor Ub (carrying the K48R substitution to prevent longer

chain formation, and optionally the K29R substitution as indicated) and 250 μM acceptor Ub (D77 variants with an additional Asp at the C-terminus to prevent usage as donor, along with other substitutions as indicated) with 20 μM UBE2R2 and 1 μM E1 in 25 mM HEPES pH 7.5, 150 mM NaCl, 2.5 mM MgCl$_2$, 12.5 mM ATP and 1 mM DTT at 37 °C overnight. Products were isolated using cation-exchange chromatography and size-exclusion chromatography in 25 mM HEPES, pH 7.5, and 150 mM NaCl (also with 1 mM TCEP for K29C variant). These di-Ubs are referred to in following sections as Ub$^{K48R}$–Ub$^{D77}$, with the distal Ub listed first, followed by the proximal (and principal acceptor) Ub.

### Fluorescent labeling

To generate fluorescently labeled donor Ub for pulse–chase assays, Cys–Ub K0 (with all lysines mutated to arginine to prevent chain formation) was purified as described above. Purified protein was preincubated for 15 min on ice with 1 mM DTT to fully reduce the N-terminal cysteine, followed by desalting twice using PD10 (GE Healthcare) desalting columns to remove reducing agents. Desalted Ub was then incubated with fivefold molar excess of fluorescein-5-maleimide (AnaSpec) at room temperature for 2 h. Products were again desalted twice and subjected to size-exclusion chromatography in 25 mM HEPES, pH 7.5, and 150 mM NaCl to remove excess dye.

### Generation of stable complexes mimicking polyubiquitylation by TRIP12

Specialized chemical-biology tools were used to covalently trap TRIP12 with its catalytic Cys, and donor and acceptor Ubs were stably linked to mimic the transition state in Ub transfer (Fig. 2a). Their basic building block, Ub–BmDPA (6×His–Ub$^{1-75}$–(E)-3-(2-(bromomethyl)-1,3-dioxolan-2-yl)prop-2-en-1-amine), was synthesized as described previously[9]. In brief, 6×His–Ub$^{1-75}$–intein–chitin binding domain (CBD) was expressed in E. coli and purified through Ni-NTA affinity chromatography. Next, the split intein fragment was thiolytically cleaved using sodium 2-mercaptoethane sulfonate (MESNa) to form 6×His–Ub$^{1-75}$–MESNa, which was further purified using size-exclusion chromatography (25 mM MES, pH 6.2, and 100 mM NaCl). Then, 6×His–Ub$^{1-75}$–MESNa (5 g L$^{-1}$) was conjugated to BmDPA (ChiroBlock, 0.4 M) in the presence of 10 mM N-hydroxysuccinimide at 30 °C and 300 rpm overnight. The resulting Ub–BmDPA species was subsequently deprotected using 40 mM p-toluenesulfonic acid in 54% (vol/vol) trifluoroacetic acid. It was then precipitated and washed with cold diethyl ether, and refolded to yield Ub–ABPO (6×His–Ub–(E)-5-amino-1-bromopent-3-en-2-one) as the final product. Reactive probes were formed by prereducing 100 μM Ub$^{K29C}$ (for K29-linked chain formation probe) or Ub$^{K48R}$–Ub$^{K29C D77}$ (for K29/K48-linked branched chain formation probe, synthesis described above) with 1 mM TCEP. The mixture was then desalted (Zeba Spin Desalting Column, 3 K MWCO) and combined with an excess of Ub–ABPO (optimized for each batch in small-scale test reactions), which was then incubated at 30 °C for 1 h. Products were isolated using Ni-NTA affinity chromatography followed by size-exclusion chromatography in 25 mM HEPES, pH 7.5, and 150 mM NaCl.

Chemically stable mimics of ubiquitylation complexes were prepared by incubating freshly purified TRIP12 (full-length or ΔN) at 2 μM with excess of reactive probe (10-fold for K29/K48-linked branched chain formation, 15-fold for K29-linked chain formation) in 25 mM HEPES, pH 7.5, 150 mM NaCl, and 1 mM TCEP at 30 °C for 1 h. Reactions were quenched by 1:1 dilution in 25 mM HEPES, pH 7.5, 150 mM NaCl, and 5 mM DTT, and were immediately plunge-frozen to prepare samples for cryo-EM.

### Cryo-EM

**Sample preparation.** CHAPSO was added to samples at a final concentration of 4 mM right before plunge-freezing. Four microliters of sample was applied on glow-discharged R1.2/1.3 200 mesh holey carbon grids (Quantifoil) and plunge-frozen in liquid ethane on a Vitrobot Mark IV (Thermo Fisher Scientific; blot force 3, blot time 3 s, 4 °C, 100% humidity).

**Data collection.** Grids were screened using SerialEM v4.1 on a Talos Arctica cryo-TEM (FEI), operated at 200 kV and equipped with a K3 direct electron detector (Gatan), to select grids suitable for high-resolution data collection. Screening datasets for TRIP12$^{ΔN}$ samples were collected at a magnification of ×22,000 and pixel size 1.841 Å px$^{-1}$, in a defocus range of −1 to −2.6 μm, and with a dose of 60.0 e$^-$ per Å$^2$ fractionated over 40 frames.

Final high-resolution datasets were collected using SerialEM v4.1 on a Titan Krios cryo-TEM (FEI) operated at 300 kV, equipped with a Bio Quantum post-column energy filter (Gatan; 10 eV) and K3 direct electron detector (Gatan) in counting mode, at a magnification of ×105,000 and pixel size 0.8512 Å px$^{-1}$. Defocus was varied from −0.6 to −2.2 μm. Samples were exposed for 3 s. Average total exposure doses were 76.5 e$^-$ per Å$^2$ for the full-length TRIP12 branched chain formation dataset, 64.8 e$^-$ per Å$^2$ for the TRIP12$^{ΔN}$ branched chain formation dataset, and 66.8 e$^-$ per Å$^2$ for the TRIP12$^{ΔN}$ chain formation dataset, fractionated over 30 frames. Detailed data collection, refinement and validation statistics are summarized in Table 1.

**Data processing.** For the dataset with full-length TRIP12 representing forging of a K29/K48-linked branched chain, dose weighting, motion correction and CTF estimation were performed in RELION 5.0 (ref. [64]) using the built-in implementation of MotionCor2 and CTFFIND 4.1 (ref. [65]), respectively. Particles were picked using Gautomatch (K. Zhang, MRC Laboratory of Molecular Biology) and extracted in RELION (4× binned, box size 80 px, applies to all datasets), then imported into cryoSPARC v4.5 (ref. [66]) for further 2D and 3D classification, re-extraction at full box size (320 px for all datasets), and high-resolution refinement. A detailed processing schematic is provided in Extended Data Fig. 2. The data obtained with full-length TRIP12 exhibited substantial orientation bias (see also Extended Data Fig. 2c), resulting in anisotropy of the final reconstruction, especially at higher resolution, which hindered accurate building of atomic models. Use of TRIP12$^{ΔN}$ largely eliminated this orientation preference (Extended Data Figs. 4c and 5c and Supplementary Video 1).

Raw movies of all TRIP12$^{ΔN}$ datasets were processed in cryoSPARC v4.5, using Patch Motion correction followed by Patch CTF estimation. For screening datasets, particles were picked using blob picker and processed as displayed in Extended Data Fig. 3. Maps obtained from screening datasets were used as reference volumes for heterogeneous refinements in the high-resolution datasets. For the high-resolution TRIP12$^{ΔN}$ K29/K48-linked branched chain formation dataset, particles were picked using templates generated with the map obtained from the screening dataset, and classified and refined as depicted in Extended Data Fig. 4. Three-dimensional variability analysis (as shown in Supplementary Video 2) was performed on the particle set before 3D classification, filtering the resolution to 6 Å and solving for 6 modes. To highlight the similarities and variations in conformations in Supplementary Video 2, maps were fit with the following individual units extracted from the final refined coordinate file: (1) the TRIP12 ARM domain, (2) the TRIP12 HEL-UBL domain and HECT domain N-lobe, (3) the distal Ub, and (4) the TRIP12 HECT domain C-lobe, C-terminus, and acceptor and donor Ubs.

For the dataset representing TRIP12$^{ΔN}$ producing a K29-linked Ub chain, particles were picked using blob picker (subsequent template picking did not yield an improved particle set) and processed as outlined in Extended Data Fig. 5. Further 3D classification and 3D variability analysis steps, beyond those given in processing schemes, were explored, but did not yield an improved resolution or reconstruction.

Maps were sharpened using DeepEMhancer (version 2020.09.07)[67]. Initial models were built based on Alphafold2 multimer[68] predictions of

TRIP12 in complex with Ub (which was placed as donor Ub and around the distal Ub binding site, both in 'loop-out' conformation). Because the dataset representing TRIP12$^{\Delta N}$ building a K29-linked chain produced the highest-resolution map, especially in the active site, this structure was used as a starting model for building and refinement of the structure representing branched chain formation. Because density for distal Ub was lower resolution and visualized at lower contour, this Ub was largely rigid-body docked in UCSF ChimeraX v1.8. Densities for the K48 linkage in the di-Ub acceptor and for the probe molecule in the active site were absent or visible only at a very low threshold in the deepEMhancer-sharpened map, but could be modeled on the basis of the unsharpened map. Initial models were refined using iterative cycles of manual fitting in Coot[69] and real-space refinement in Phenix[70]. Structural renderings were created using UCSF ChimeraX v1.8.

**Biochemical assays.** Pulse–chase assays were performed to track the transfer of fluorescently labeled donor Ub (*Ub(K0)) from E2 to E3 enzymes and further onto substrate. UBE2L3 was used as the E2 because a corresponding activity-based probe showed higher reactivity toward TRIP12 than did a UBE2D2-derived probe in a previous study[71].

For experiments assaying activity of TRIP12, unless indicated otherwise, 7.5 µM fluorescently labeled Ub K0 (henceforth referred to as *Ub(K0)) and 6 µM UBE2L3 were incubated with 0.3 µM UBA1 in 50 mM HEPES, pH 7.5, 100 mM NaCl, 2.5 mM MgCl$_2$, 1.5 mM ATP, and BSA (0.05 g L$^{-1}$) at room temperature for 30 min. The pulse reaction was quenched by 1:3 dilution in 50 mM HEPES, pH 7.5, 50 mM NaCl, and 50 mM EDTA and incubation at room temperature for 5 min. Chase reactions were then initiated by mixing 200 nM UBE2L3-*Ub(K0) with 200 nM TRIP12 (or TRIP12 variants) and acceptor. Unless noted otherwise, WT full-length TRIP12 was used for assays. Specific acceptors and their concentrations for each experiment are reported below, in order of appearance in the main text. Nomenclature for mutant K48-linked di-Ubs is delineated in the section describing their generation. Where no mutations are indicated, native K48 di-Ub from in vitro K48 chain formation reactions was used (see previous section).

Determination of substrate di-Ub linkage specificity (Fig. 1a) was performed using 200 nM or 1 µM of mono-Ub or di-Ub of designated linkage types (purified from in vitro chain formation reactions or synthesized as specified in previous section) as the acceptor.

For the substrate concentration dependence assays (Extended Data Fig. 1a), K48 di-Ub and mono-Ub were added at 0.2, 0.5, 1, or 2 µM, as indicated. Negative control reactions contained the corresponding K29R variants at 2 µM.

The preference for proximal over distal Ub (Fig. 1b) was characterized by testing the activity of TRIP12 on 200 nM of Ub$^{K48R}$–Ub$^{D77}$, Ub$^{K48R}$–Ub$^{K29R D77}$, Ub$^{K29R K48R}$–Ub$^{D77}$ or Ub$^{K29R K48R}$–Ub$^{K29R D77}$.

The influence of acceptor side chain length was tested by comparing TRIP12 activity on 200 nM of semi-synthetic Ub$^{K29R K48R}$–Ub$^{K29Dap D77}$ (C1), Ub$^{K29R K48R}$–Ub$^{K29Dab D77}$ (C2), Ub$^{K29R K48R}$–Ub$^{K29Orn D77}$ (C3), Ub$^{K29R K48R}$–Ub$^{D77}$ (C4), Ub$^{K29R K48R}$–Ub$^{K29hLys D77}$ (C5), or fully recombinantly produced Ub$^{K29R K48R}$–Ub$^{D77}$ (C4 bio). Band intensities of tri-Ub products at 7-s timepoints were quantified using ImageJ[72] and plotted using GraphPad Prism 10.

The truncated TRIP12$^{\Delta N}$ variant was validated (Extended Data Fig. 1c) by comparing the activity of 200 nM TRIP12 or TRIP12$^{\Delta N}$ on 200 nM K48 di-Ub or mono-Ub, and the corresponding Ub$^{K29R}$ variants were used as negative controls.

Assays showing the loss of activity in TRIP12 point mutants (Figs. 3b and 4a–c and Extended Data Fig. 8a) were performed using 200 nM full-length TRIP12 (carrying the indicated substitutions or WT as positive control), and 200 nM K48 di-Ub or 1 µM mono-Ub as the acceptor.

To test the effects of substitutions in a K48 di-Ub acceptor (Figs. 3b and 4b), 200 nM of Ub$^{K29R K48R}$–Ub$^{D77}$ ('WT' in legend), Ub$^{K29R K48R}$–Ub$^{R42A D77}$, Ub$^{K29R K48R}$–Ub$^{R42E D77}$, Ub$^{K29R K48R}$–Ub$^{E16K D77}$, Ub$^{K29R K48R}$–Ub$^{D21A D77}$ or Ub$^{K29R K48R}$–Ub$^{N25D D77}$ was added to chase reactions, together with 200 nM WT

TRIP12. For substitutions in the mono-Ub acceptor (Extended Data Fig. 8a), 1 µM Ub–6×His or the corresponding variants were used instead.

In assays testing TRIP12 activity on mono-, K29 tetra-, and K48 tetra-Ub with different equivalents of UBE2L3-*Ub(K0) (Extended Data Fig. 9a), two pulse reactions with 15 µM or 75 µM *Ub(K0) and 12 µM or 60 µM UBE2L3 (for 1 and 4 equivalents, respectively) were set up and further processed as described above, such that final assay concentrations were 1 µM or 4 µM UBE2L3-Ub; 1 µM of mono-Ub or K29 tetra-Ub or 250 nM K48 tetra-Ub was used as the acceptor in chase reactions.

Assays comparing K29R and WT donor in the modification of mono-, K29 tetra-, and K48 tetra-Ub by TRIP12 (Extended Data Fig. 9b) were performed using fourfold excess of UBE2L3-Ub. Pulse reactions were performed as described above, but using the indicated untagged Ub variants (K29R or wild type) as donors. Acceptors were added to chase reactions at a final concentration of 1 µM for mono- and K29 tetra-Ub or 250 nM for K48 tetra-Ub.

For the DUB cleavage assay to distinguish products on the basis of susceptibility to OTUB1* (a K48-linkage-specific DUB), both alone and in combination with TRABID (a K29-linkage-specific DUB) (Extended Data Fig. 9c), reactions were set up with the indicated acceptor, consisting of mono-Ub (1 µM), K29-linked tetra-Ub (1 µM), and K48-linked chains (250 nM), including 4 equivalents of UBE2L3-Ub (untagged wild type), as described in previous section. The mixtures were incubated for 5 min, then quenched by addition of 500 mM HEPES, pH 7.5, 1.5 M NaCl, and 50 mM DTT (10 µl in 80 µl reaction). Reactions were split into three 20-µl portions, and 2.5 µl of OTUB1* (1 µM final concentration) or DUB dilution buffer (50 mM HEPES, pH 7.5, 150 mM NaCl, and 10 mM DTT) were added. Reactions were incubated at 37 °C for 30 min, followed by the addition of 2.5 µl TRABID (500 nM final concentration) or DUB dilution buffer and further incubation at 37 °C, then were quenched by the addition of reducing SDS–PAGE sample buffer.

In assays comparing mutants that distinguish chain formation by different linkage-specific HECT E3s (Fig. 5d, left), chase reactions contained 200 nM TRIP12 and 1 µM Ub–6×His, Ub$^{A46F}$–6×His, or Ub$^{D21A}$–6×His. Corresponding experiments with UBR5 (Fig. 5d, right) were performed using 200 nM UBE2D2-*Ub (as E2-Ub, generated as described in ref. 9), 200 nM UBR5, and 2 µM Ub-6×His or variants.

Samples were collected at the indicated timepoints and quenched by mixing with SDS–PAGE sample buffer (final concentration: 50 mM Tris, pH 6.8, 10% glycerol, 15 mM EDTA, 0.5% bromophenol blue, 2% SDS; for reducing, 100 mM DTT was added), then analyzed by SDS–PAGE on 6–22% gels. Gels of assays using fluorescent *Ub(K0) as the donor were scanned using an Amersham Typhoon 5 (Cy2 channel), followed by staining with Coomassie Brilliant Blue to visualize protein inputs. Gels of assays containing untagged WT and K29R donor Ubs were subjected to immunoblotting using anti-Ub P4D1 antibody (Cell Signaling Technology, cat. no. 14049S, diluted 1:5,000 in 5% milk in TBS-T), developed using Pierce ECL substrate (Thermo Scientific, cat. no. 32209), and imaged on an Amersham Imager 600 (GE Lifesciences). Blots were denatured after transfer by incubation in 6 M guanidinium hydrochloride for 30 min. All displayed gels and blots are representative of at least two independent technical replicates.

### Reporting summary

Further information on research design is available in the Nature Portfolio Reporting Summary linked to this article.

### Data availability

Methods for synthesizing ubiquitins are provided in Supplementary Note. Raw images are provided as source data. Structural data are available from EMDB and RCSB: TRIP12$^{\Delta N}$ branched K29/K48-linked chain formation: EMD-51429, PDB 9GKM; TRIP12$^{\Delta N}$ K29-linked di-ubiquitin formation: EMD-51430, PDB 9GKN; full-length TRIP12 branched K29/K48-linked chain formation: EMD-51428. Source data are provided with this paper.

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

## Acknowledgements

We thank D. Bollschweiler and T. Schäfer of the Max Planck Institute of Biochemistry Cryo-EM Facility (RRID:SCR_025744) as well as R. Prabu for assistance with cryo-EM; G. Kleiger, C. Klose, J. Farnung, and L. Kiss for scholarly discussions and advice; S. von Gronau for baculovirus and insect cell culture; L. Kiss for providing OTUB1* and K29-linked chains; and other members of the Schulman lab for advice and assistance, especially J. Kellermann, S. Sepic, E. Papadopoulou, and K. Gottemukkala. This study has received the following funding: to S.A.M., a Ph.D. fellowship from the Boehringer Ingelheim Fonds; to M.P.C.M., NWO (VIDI Grant VI. 213.110); to B.A.S., Max Planck Gesellschaft, the European Research Council (ERC) under the European Union's Horizon 2020 research and innovation programme (grant agreement no. 789016-NEDD8Activate), and the framework of SFB 1035 (German Research Foundation DFG, Sonderforschungsbereich 1035, Projektnummer 201302640, project A13).

## Author contributions

Biochemistry: S.A.M.; cryo-EM: S.A.M.; structure building and refinement: S.A.M.; protein purification: S.A.M., L.A.S., R.V., J.L. and K.N.S.; preparation of chemically-modified Ubs and chains: M.D.M., M.P.C.M. and S.A.M.; data analysis: S.A.M. and B.A.S.; supervision: B.A.S.; paper preparation: S.A.M. and B.A.S., with input from all authors.

## Funding

## Competing interests

B.A.S. is a member of the scientific advisory board of Proxygen, has served on the scientific advisory boards of Interline and Biotheryx, and is a coinventor of intellectual property licensed to Cinsano. The other authors declare no competing interests.

## Additional information

**Extended data** is available for this paper at https://doi.org/10.1038/s41594-025-01561-1.

**Correspondence and requests for materials** should be addressed to Brenda A. Schulman.

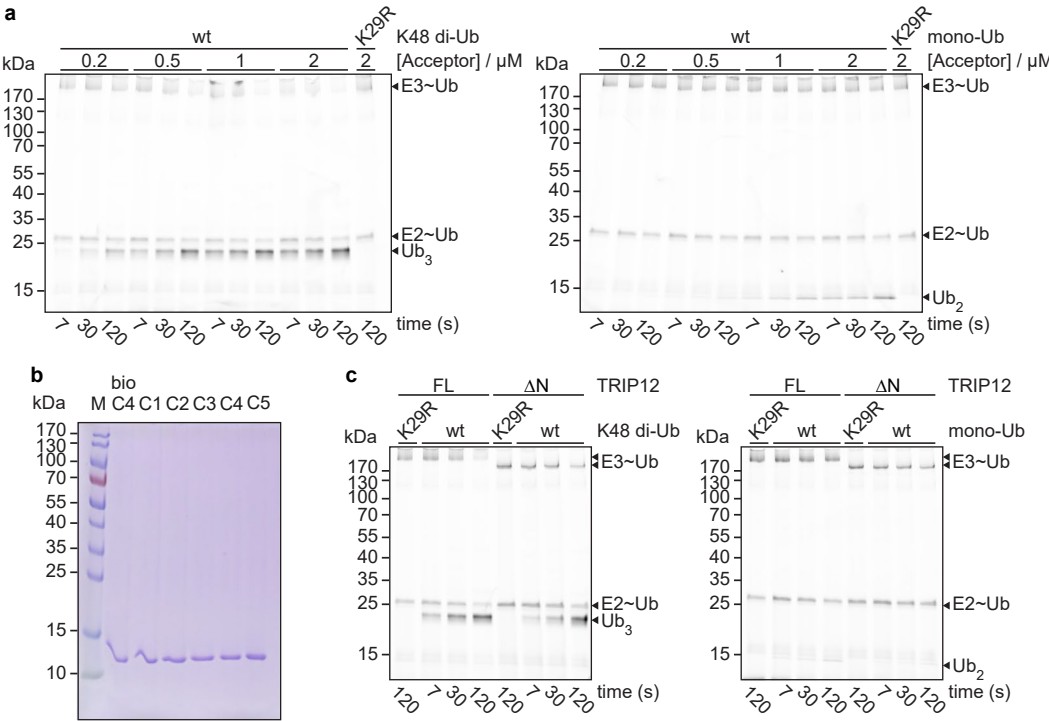

**Extended Data Fig. 1 | Defining features and elements of linkage-specific ubiquitin chain formation by TRIP12. a**, Assay comparing TRIP12 activity toward K48-linked di-Ub and mono-Ub acceptors. Experiment was performed in pulse-chase format, where the E2-Ub intermediate formed in the pulse-reaction was UBE2L3 thioester-bonded to lysineless (K0) Ub labeled with a fluorescent N-terminal tag. The chase was initiated by adding TRIP12 together with mono-Ub or K48 di-Ub acceptor, or corresponding K29R variants, at indicated concentrations. Fluorescent Ub is tracked through the cascade over time by migration in non-reducing SDS-PAGE, and detected by fluorescent scan. Products are denoted $Ub_2$ and $Ub_3$ for reactions adding a Ub to mono- or di-Ub acceptors, respectively. Results show TRIP12 activity towards a K48-linked di-Ub target at relatively lower concentrations than mono-Ub. Experiments comparing K29R substituted K48-linked di-Ub and mono-Ub acceptors were performed at maximum concentration (n = 3 independent technical replicates). **b**, Coomassie-stained gel showing purified semi-synthetic di-Ubs with position 29 of the proximal Ub differing by number of methylene groups – one (C1), two (C2), three (C3), four (C4) or five (C5) - between the alpha carbon and side-chain amino group: L-2,3-diaminopropionic acid (Dap) has one methylene group in the analog replacing native K29; L-2,4-diaminobutyric acid (Dab) has two methylene groups in the analog replacing K29; L-ornithine (Orn) has three methylene groups in the analog replacing K29; the native side-chain L-lysine (Lys) has four methylene groups; and L-homolysine (hLys) has five methylene groups in the analog replacing K29. C4bio refers to Ub with native Lys, obtained through recombinant expression in E. coli. **c**, Assay as in a, testing ubiquitylation activity of truncated TRIP12$^{\Delta N}$ versus WT TRIP12 towards K48 di- and mono-Ub, both at 200 nM (n = 2 independent technical replicates).

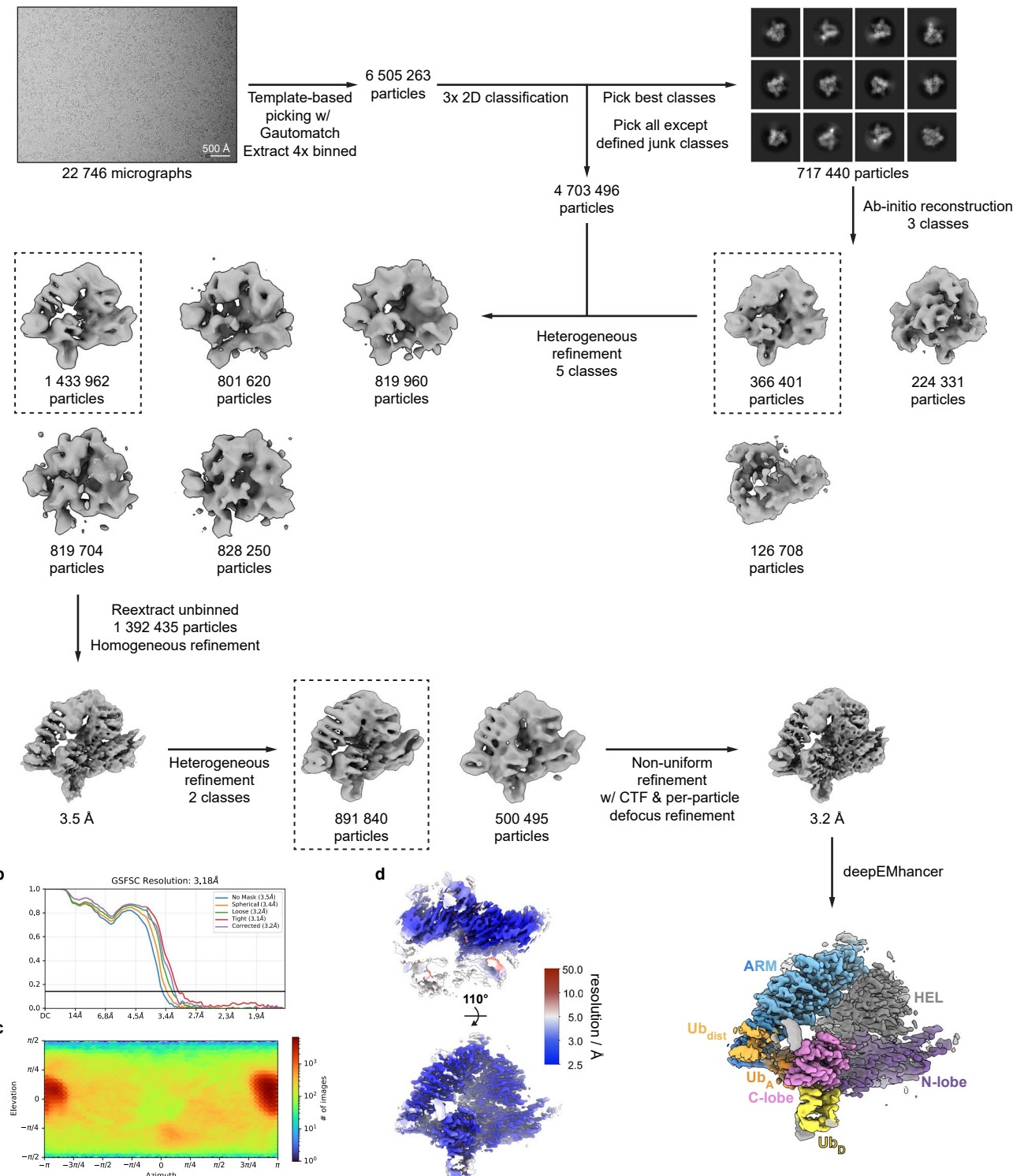

**Extended Data Fig. 2 | Cryo-EM processing of structure representing full-length TRIP12 forging a branched K29/K48-linked ubiquitin chain.** **a**, Processing scheme. Scalebar on micrograph corresponds to 500 Å. Data processed in Cryosparc 4.5.3 ultimately yielded a 3D reconstruction with a resolution of 3.2 Å by the gold-standard Fourier shell correlation of 0.143.

**b**, Gold-standard Fourier shell correlation plot. **c**, Orientation distribution plot of particles from final refinement. **d**, Local resolution estimation projected onto deepEMhancer-sharpened map. This showcases the heterogeneous quality of the map, with relatively lower resolution for key entities including the donor Ub and K48-linked acceptor.

**a** TRIP12$^{\Delta N}$ - K29/K48-linked branched ubiquitin chain formation - screening dataset

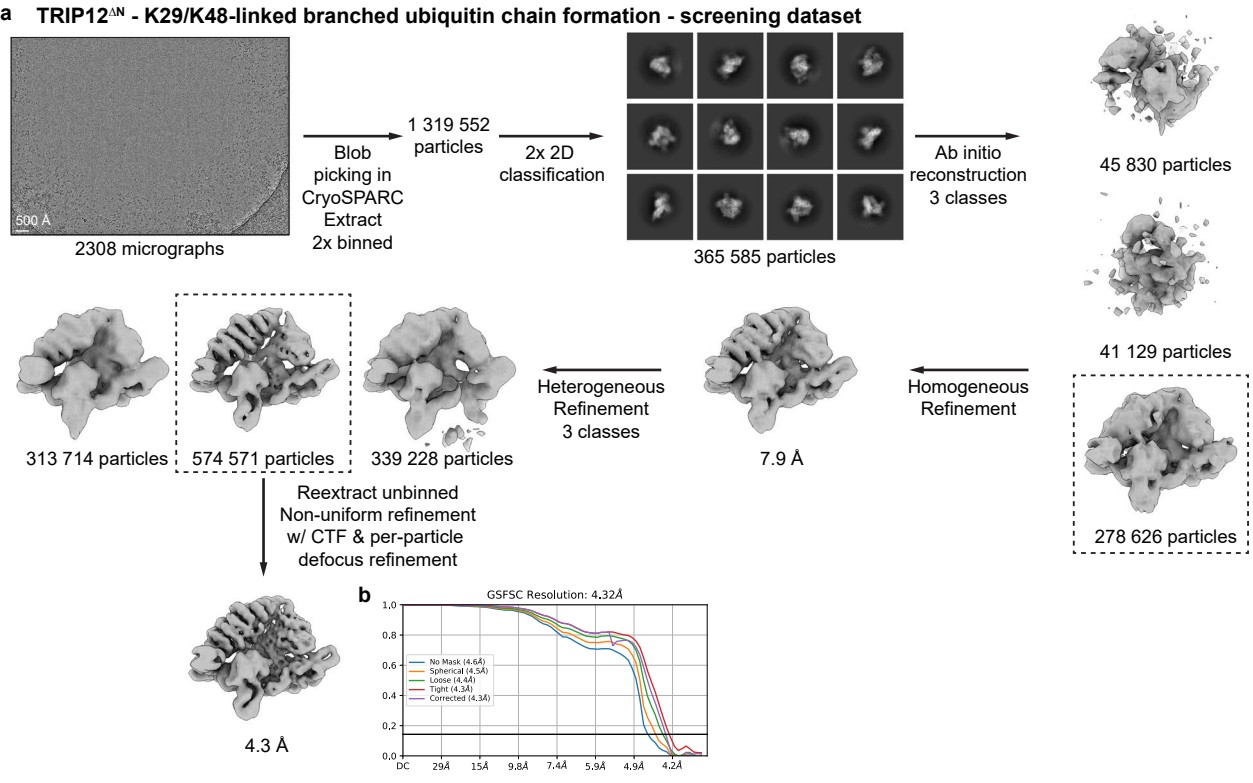

**c** TRIP12$^{\Delta N}$ - K29-linked chain formation - screening dataset

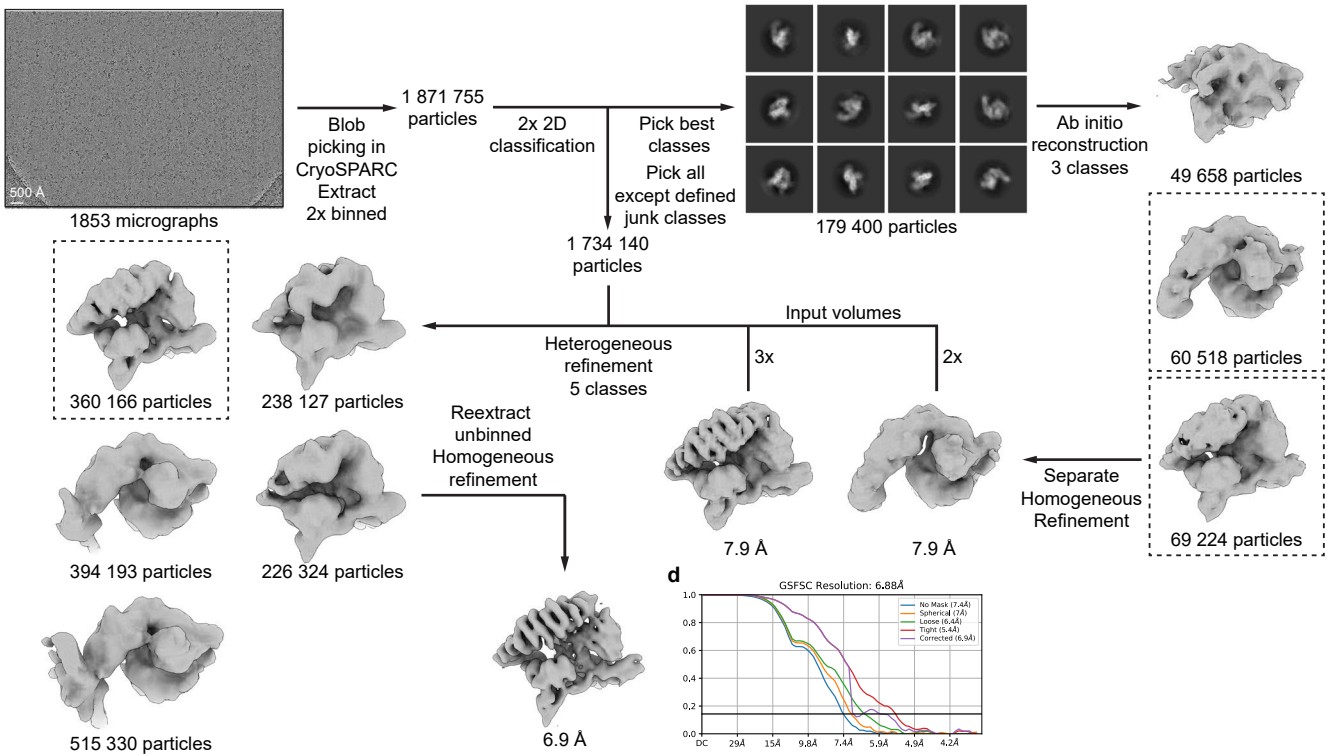

**Extended Data Fig. 3 | Cryo-EM processing schemes for TRIP12$^{\Delta N}$ screening datasets. a**, Processing scheme for screening dataset of TRIP12$^{\Delta N}$ forming a branched K29/K48-linked ubiquitin chain. Final volume was used as input for Heterogeneous Refinement in Extended Data Fig. 4. Scalebar on micrograph corresponds to 500 Å. Data processed in Cryosparc 4.5.3 yielded a 3D reconstruction with a resolution of 4.3 Å by the gold-standard Fourier shell correlation of 0.143. **b**, Gold-standard Fourier shell correlation plot for **a**. **c**, Processing scheme for screening dataset of TRIP12$^{\Delta N}$ forging a K29-linked di-ubiquitin chain. Final volume was used as input for Heterogeneous Refinement in Extended Data Fig. 5. Scalebar on micrograph corresponds to 500 Å. Data processed in Cryosparc 4.5.3 yielded a 3D reconstruction with a resolution of 6.9 Å by the gold-standard Fourier shell correlation of 0.143.

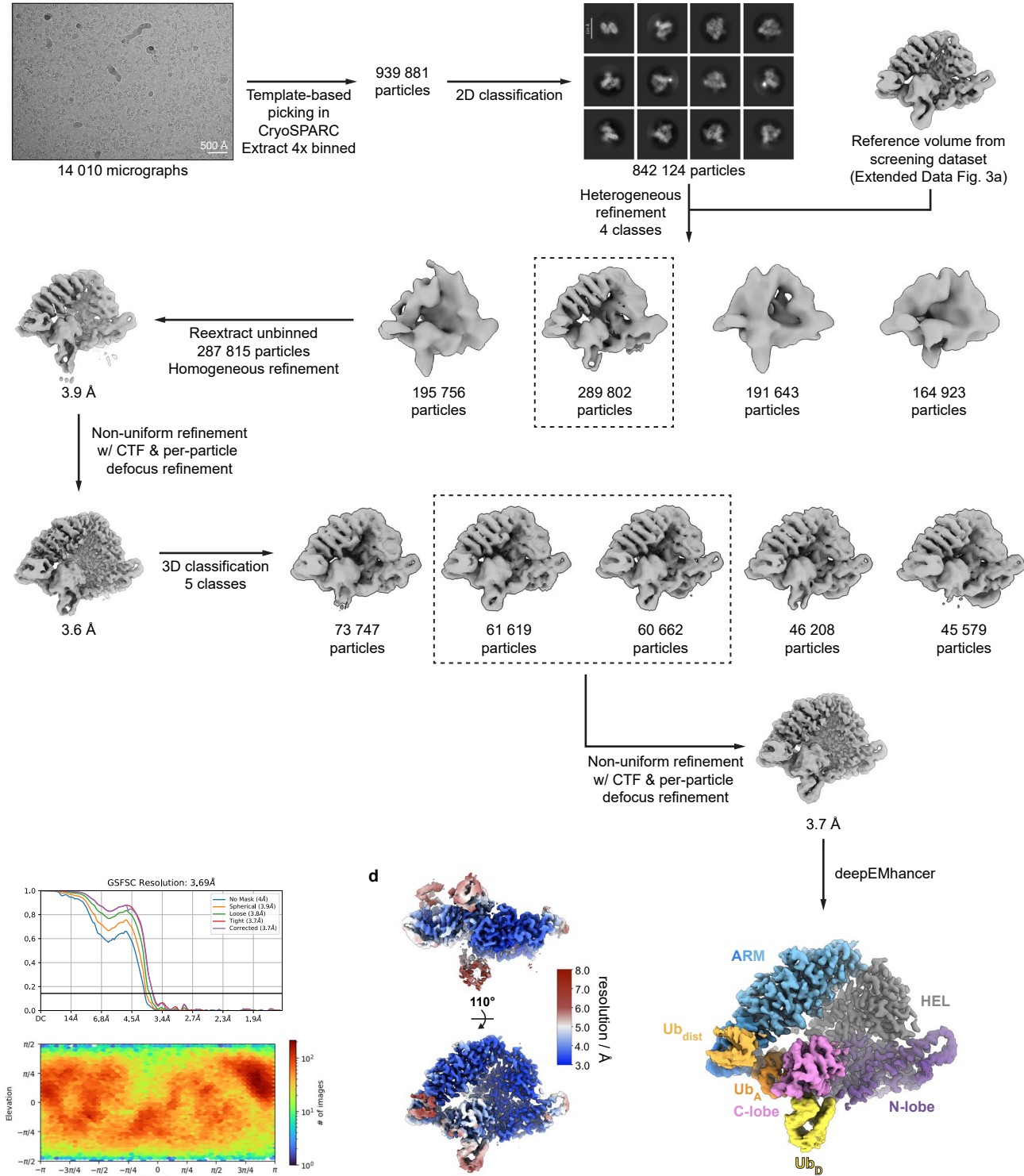

**a**  TRIP12$^{\Delta N}$ - K29/K48-linked branched ubiquitin chain formation

**Extended Data Fig. 4 | Cryo-EM processing scheme from structure representing TRIP12$^{\Delta N}$ forming a branched K29/K48-linked ubiquitin chain.** **a**, Processing scheme. Scalebar on micrograph corresponds to 500 Å. Data processed in Cryosparc 4.5.3 yielded a 3D reconstruction with a resolution of 3.7 Å by the gold-standard Fourier shell correlation of 0.143. **b**, Gold-standard Fourier shell correlation plot. **c**, Orientation distribution plot of particles from final refinement. **d**, Local resolution estimation projected onto deepEMhancer-sharpened map.

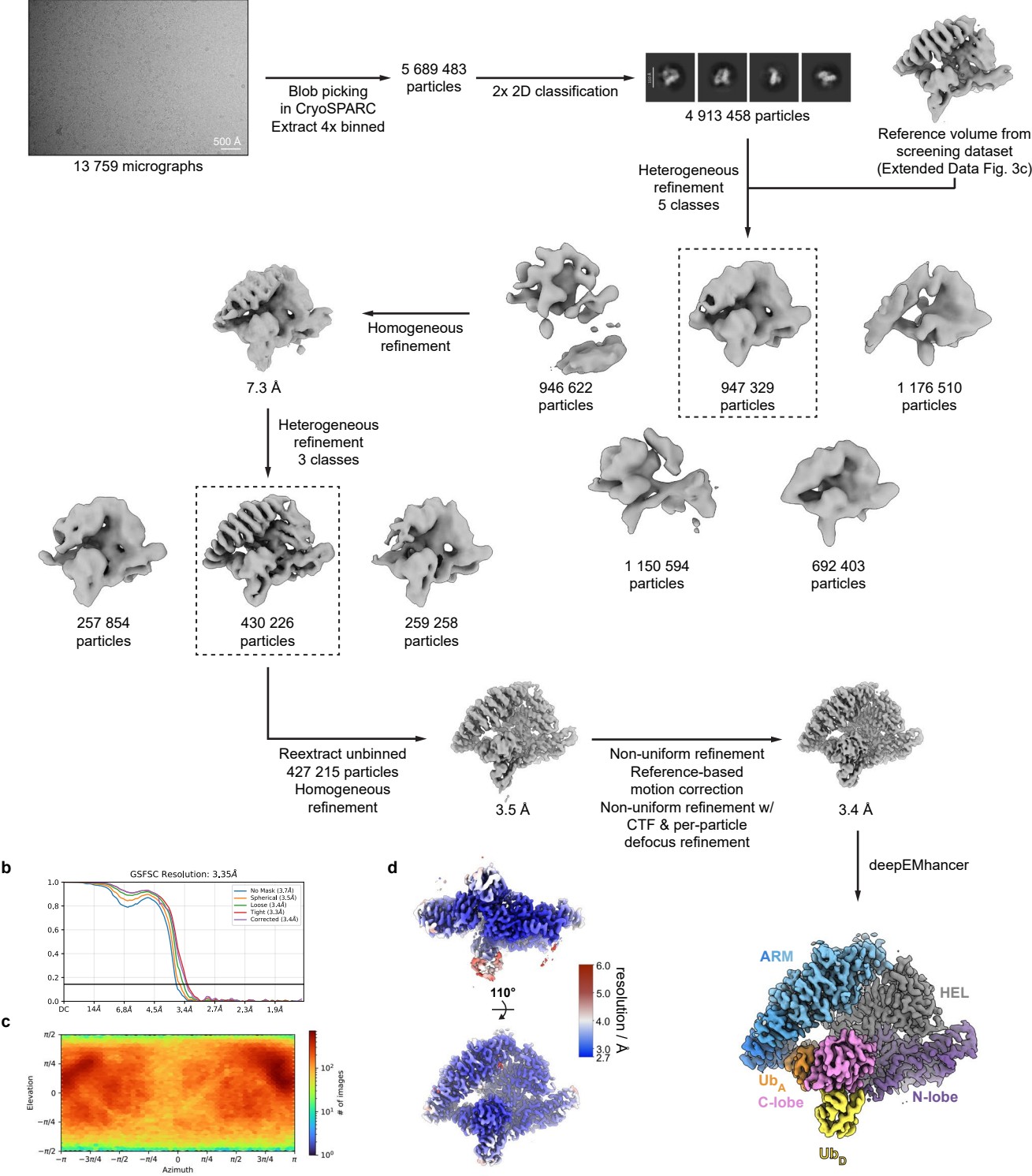

**a**

**TRIP12$^{\Delta N}$ - K29-linked chain formation**

**Extended Data Fig. 5 | Cryo-EM processing scheme from structure representing TRIP12$^{\Delta N}$ forging a K29-linked di-ubiquitin chain. a**, Processing scheme. Scalebar on micrograph corresponds to 500 Å. Data processed in Cryosparc 4.5.3 yielded a 3D reconstruction with a resolution of 3.4 Å by the gold-standard Fourier shell correlation of 0.143. **b**, Gold-standard Fourier shell correlation plot. **c**, Orientation distribution plot of particles from final refinement. **d**, Local resolution estimation projected onto deepEMhancer-sharpened map.

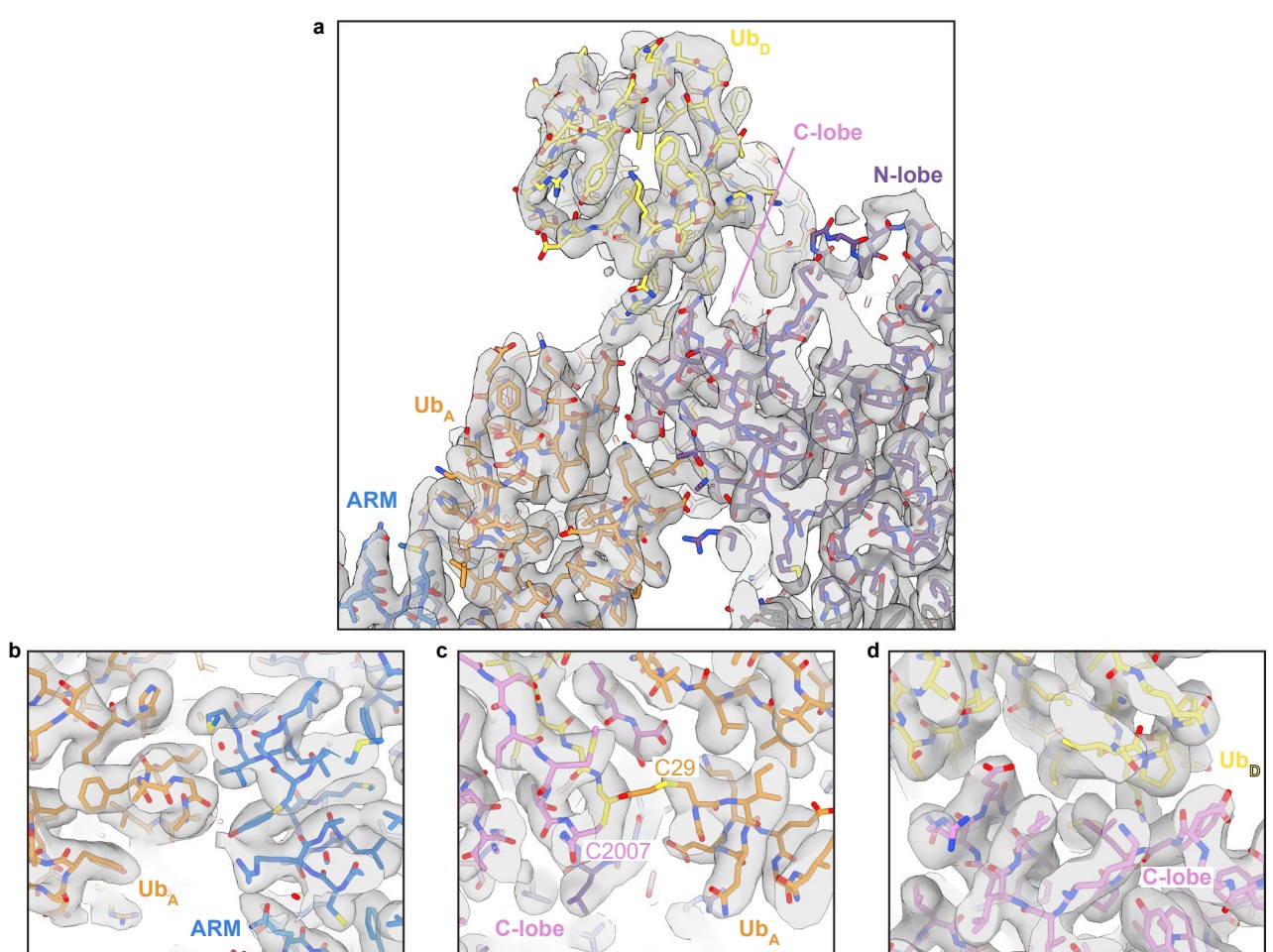

**Extended Data Fig. 6 | Example fits of model to cryo-EM map over key interactions.** Atomic model representing intermediate in TRIP12$^{\Delta N}$ forming a K29 linkage to Ub, shown in sticks representation and fit in corresponding deepEMhancer-sharpened map. **a**, Overview of catalytic assembly between acceptor and donor Ubs (orange and yellow, respectively), and TRIP12 ARM (blue) and HECT (C-lobe violet, N-lobe purple) domains. **b**, Close-up of interface between acceptor Ub and TRIP12 ARM domain. **c**, Close-up of active site interface between acceptor Ub and TRIP12 C-lobe. **d**, Close-up of interface between donor Ub and TRIP12 C-lobe.

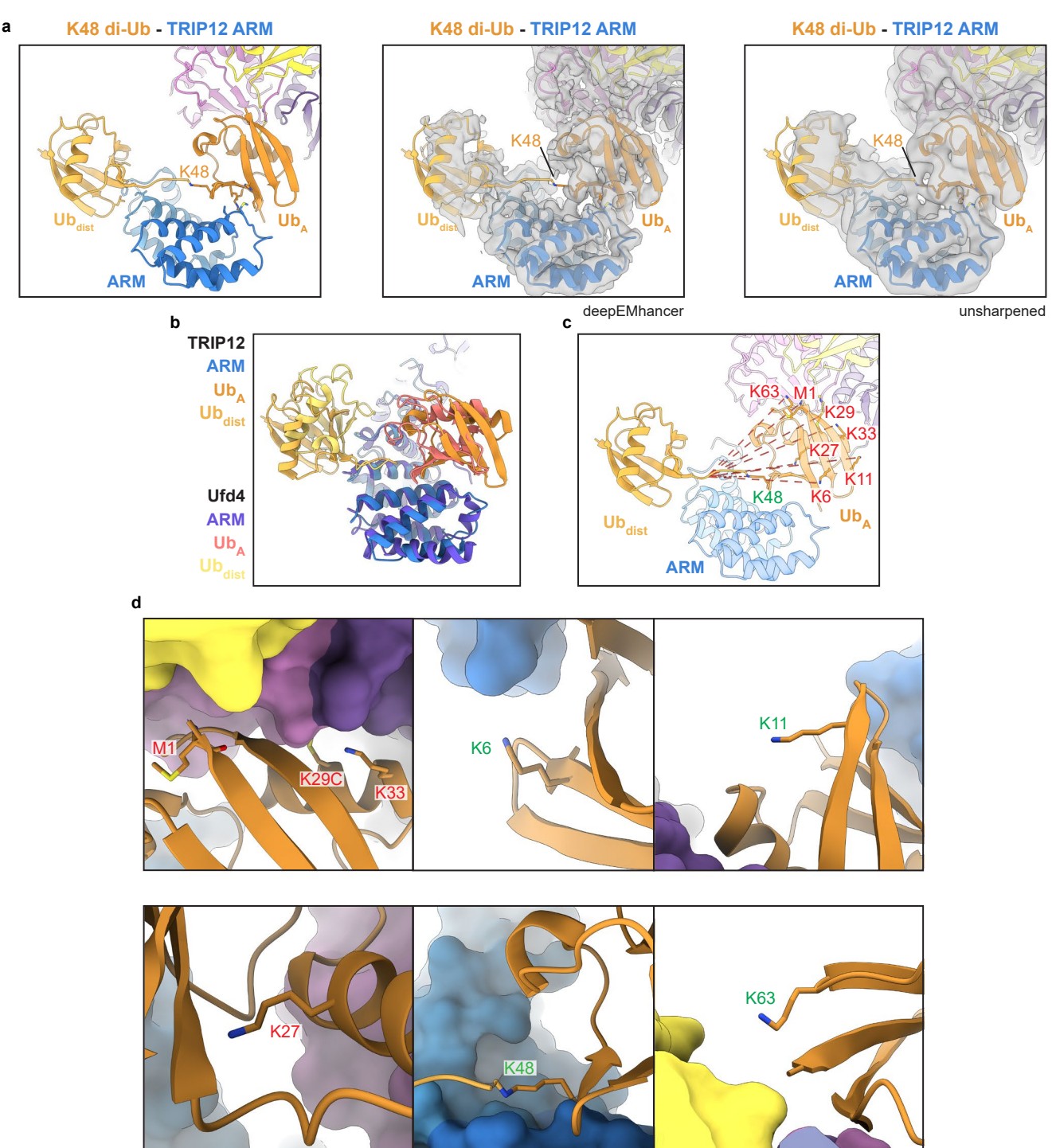

**Extended Data Fig. 7 | Structural basis for TRIP12's preference for linking ubiquitin to K29 in the proximal ubiquitin of a K48-linked chain. a**, Close-ups of structure representing K29/K48-linked branched chain formation, showing TRIP12 tandem Ub binding regions in ARM domain recognize two Ubs in geometry specialized for K48 linkage. Left, model only; center, model in deepEMhancer-sharpened map; right, model in unsharpened map, showing density for K48 linkage and distal Ub C-term that is lost upon sharpening. **b**, Comparison of tandem acceptor K48 di-Ub binding between TRIP12 and Ufd4 (PDB 8J1P), aligned on ARM domain. **c**, Close-up of structure representing K29/K48-linked branched chain formation, showing K48-linked di-Ub bound to ARM domain and positions of lysine residues and M1 on acceptor Ub, with distances to most C-terminal structured part of distal Ub indicated. No other linkage type besides K48 appears compatible with distal Ub binding and precise placement of the acceptor K29 in the active site. **d**, Close-ups of acceptor Ub Met1 and Lys residues with interfacing domains in surface representation, showing accessibility for linkage to a distal Ub C-terminus. Available residues are labeled in green, buried residues in red.

a

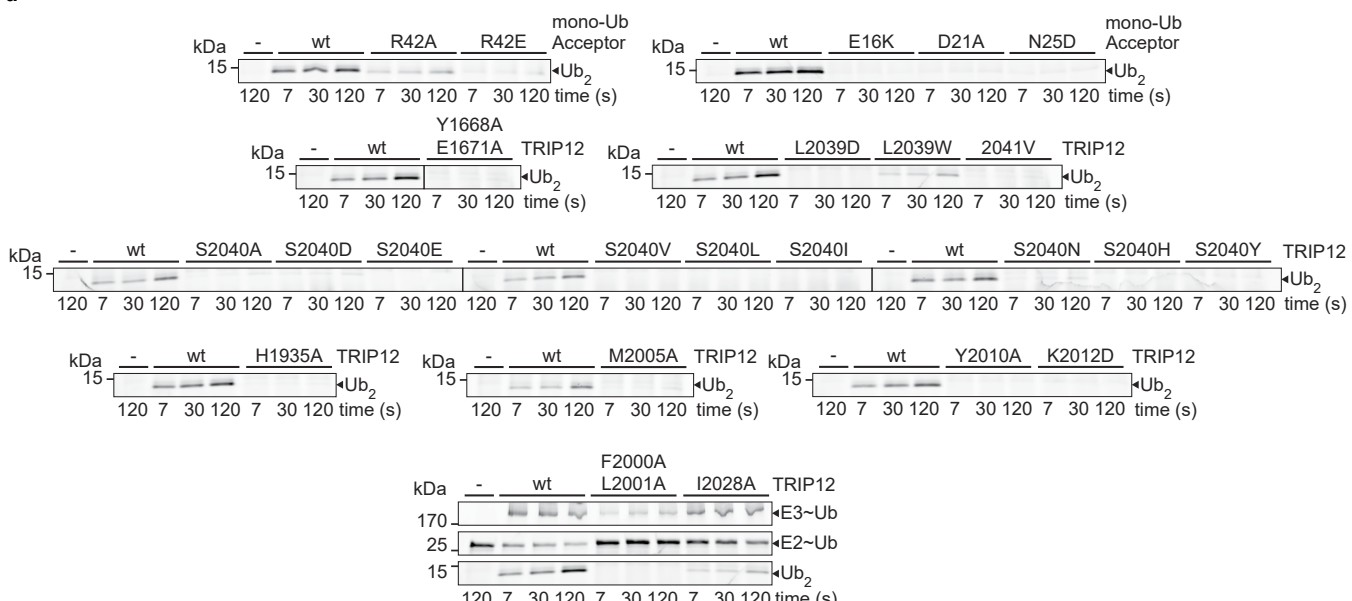

**Extended Data Fig. 8 | Effects of interface mutations on TRIP12 forging K29-linked di-Ub in assays using a mono-Ub acceptor. a**, Pulse-chase assays testing effects of indicated mutations in TRIP12 or acceptor Ub on formation of K29-linkages to a mono-Ub acceptor, producing di-Ub (denoted Ub$_2$). Acceptor was added to chase reactions at final concentration of 1 µM ($n$ = 2 independent technical replicates).

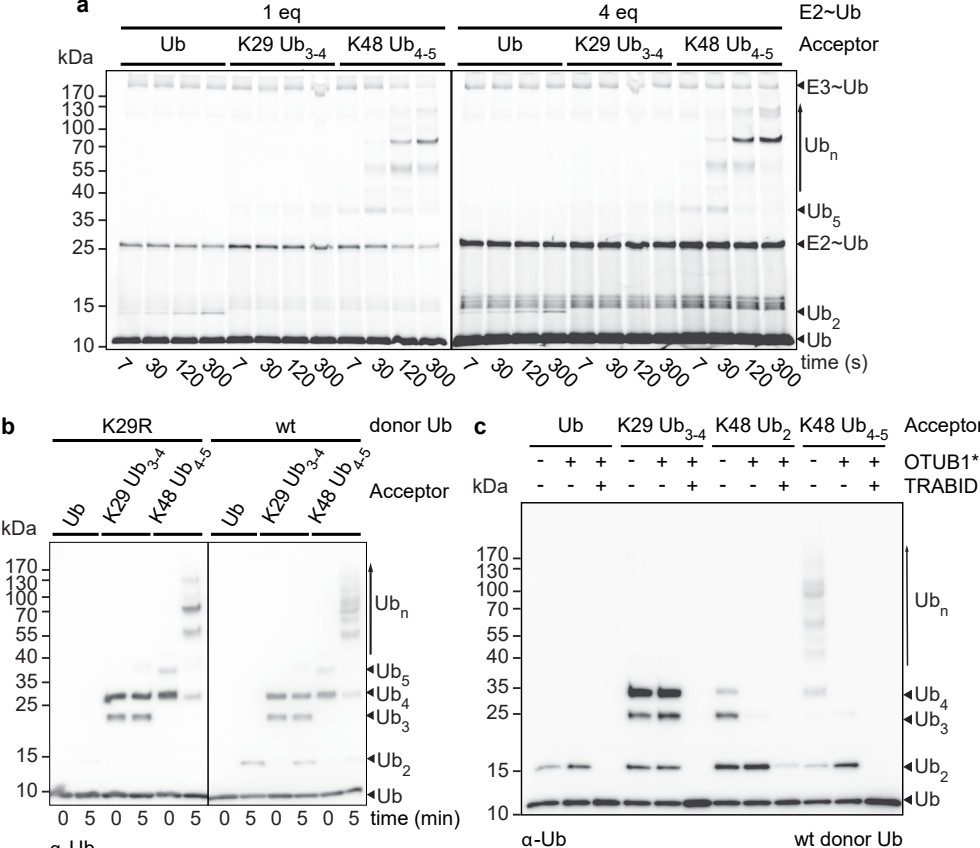

**Extended Data Fig. 9 | TRIP12 preferentially monoubiquitylates multiple Ubs within a K48 chain acceptor rather than extending a K29-linked chain.**
**a**, Assay testing TRIP12 modification of mono-Ub, K29-linked chains (primarily tetra-Ub but some tri-Ub), and K48-linked chains (primarily tetra-Ub with a minor amount of penta-Ub) with either equimolar or 4-fold excess of E2-Ub (UBE2L3 thioester-bonded to Ub(K0) with a fluorescent N-terminal tag). Products beyond penta-Ub are challenging to assign accurately due to varying migration behavior of branched species in SDS-PAGE, and are therefore collectively denoted $Ub_n$.

**b**, Immunoblot detecting Ub in products of pulse-chase assays performed with Ub, K29 tetra-Ub (with some tri-Ub) or K48 tetra-Ub (with some penta-Ub) acceptor and 4-fold molar excess of E2-Ub, using untagged K29R or WT donor Ub as indicated. **c**, Product validation by cleavage with linkage-specific DUBs, OTUB1* for K48-linked chains and TRABID for K29-linked chains. Treatment with OTUB1* collapses majority of products formed on K48 chains to di-Ub species, which are further eliminated by addition of TRABID. **a-c**, All gels are representative of $n = 2$ independent technical replicates.

**a**

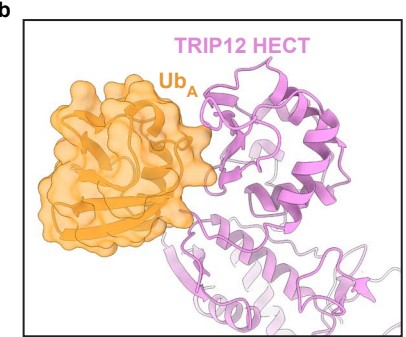

HECT C-lobe~Ub$_D$
TRIP12~Ub$_D$

UBR5~Ub$_D$ (8C07)
Rsp5~Ub$_D$ (4LCD)
HUWE1~Ub$_D$ (6XZ1)
HUWE1~Ub$_D$ (6FYH)
NEDD4L~Ub$_D$ (3JW0)
SMURF2~Ub$_D$ (6FX4)
NEDD4~Ub$_D$ (4BBN)

Ufd4~Ub$_D$ (8J1P)

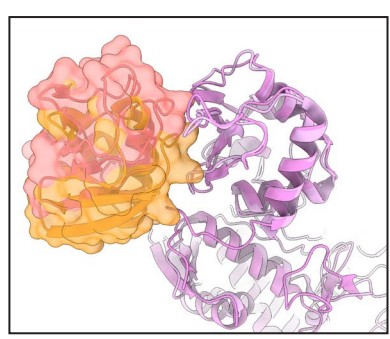

**b**

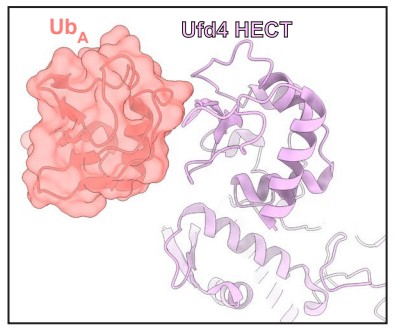

**Extended Data Fig. 10 | Comparison to prior structures shows broad conservation of TRIP12-donor Ub arrangement. a**, Superimposition of HECT C-lobe and donor Ub (Ub$_D$) from structure of TRIP12 with structures of UBR5 (PDB 8C07), Rsp5 (PDB 4LCD), HUWE1 (PDB 6XZ1, 6FYH), NEDD4 (PDB 4BBN), NEDD4L (PDB 3JW0) and SMURF2 (PDB 6FX4) (left), or with Ufd4 (PDB 8J1P, right).

Structures were aligned over their C-lobes. **b**, Comparison of positioning of the acceptor Ub (Ub$_A$) in the HECT C-lobe interface between structures for TRIP12 (left, acceptor Ub in orange) and Ufd4 (center, acceptor Ub in red), or both superimposed (right). Structures were aligned over their C-lobes.

# Reporting Summary

## Statistics

For all statistical analyses, confirm that the following items are present in the figure legend, table legend, main text, or Methods section.

| n/a | Confirmed | |
|---|---|---|
| ☐ | ☒ | The exact sample size (*n*) for each experimental group/condition, given as a discrete number and unit of measurement |
| ☐ | ☒ | A statement on whether measurements were taken from distinct samples or whether the same sample was measured repeatedly |
| ☒ | ☐ | The statistical test(s) used AND whether they are one- or two-sided *Only common tests should be described solely by name; describe more complex techniques in the Methods section.* |
| ☒ | ☐ | A description of all covariates tested |
| ☒ | ☐ | A description of any assumptions or corrections, such as tests of normality and adjustment for multiple comparisons |
| ☒ | ☐ | A full description of the statistical parameters including central tendency (e.g. means) or other basic estimates (e.g. regression coefficient) AND variation (e.g. standard deviation) or associated estimates of uncertainty (e.g. confidence intervals) |
| ☒ | ☐ | For null hypothesis testing, the test statistic (e.g. *F*, *t*, *r*) with confidence intervals, effect sizes, degrees of freedom and *P* value noted *Give P values as exact values whenever suitable.* |
| ☒ | ☐ | For Bayesian analysis, information on the choice of priors and Markov chain Monte Carlo settings |
| ☒ | ☐ | For hierarchical and complex designs, identification of the appropriate level for tests and full reporting of outcomes |
| ☒ | ☐ | Estimates of effect sizes (e.g. Cohen's *d*, Pearson's *r*), indicating how they were calculated |

*Our web collection on statistics for biologists contains articles on many of the points above.*

## Software and code

Policy information about availability of computer code

| | |
|---|---|
| Data collection | Cryo-EM data acquisition: SerialEM v4.1; Gel & blot imaging: Amersham Imager 600; Fluorescent gel scanning: Amersham Typhoon |
| Data analysis | Cryo-EM data processing: CryoSparc v4.4.0, RELION v5.0 beta, Gautomatch v0.56, CTFFIND v4.1; Structure Analysis and Visualization: ChimeraX v1.8, DeepEMhancer version 2020.09.07 (https://github.com/rsanchezgarc/deepEMhancer); Model Building & Refinement: COOT v0.9.6, Phenix.refine v1.21.1, AlphaFold2; Sequence alignment: MUSCLE (as implemented in Snapgene v7.2); Biochemical Data Analysis: ImageJ 1.54, GraphPad Prism 10; Figure Generation: Adobe Illustrator 2024, ChemDraw 22.2. All software used was either available commercially (CryoSPARC, Snapgene, GraphPad Prism, Adobe Illustrator, ChemDraw) or as open source (others). No custom code was generated or used for analyses in this study. |

For manuscripts utilizing custom algorithms or software that are central to the research but not yet described in published literature, software must be made available to editors and reviewers. We strongly encourage code deposition in a community repository (e.g. GitHub). See the Nature Portfolio guidelines for submitting code & software for further information.

## Data

Policy information about availability of data

All manuscripts must include a data availability statement. This statement should provide the following information, where applicable:
- Accession codes, unique identifiers, or web links for publicly available datasets
- A description of any restrictions on data availability
- For clinical datasets or third party data, please ensure that the statement adheres to our policy

The structural data will be available from EMDB and RCSB upon manuscript publication (TRIP12deltaN branched K29/K48-linked chain formation: EMD-51429, PDB 9GKM; TRIP12deltaN K29-linked di-ubiquitin formation: EMD-51430, PDB 9GKN). Cryo-EM map were deposited and are available via the following accession number: TRIP12 FL branched K29/K48-linked chain formation: EMD-51428.

## Research involving human participants, their data, or biological material

Policy information about studies with human participants or human data. See also policy information about sex, gender (identity/presentation), and sexual orientation and race, ethnicity and racism.

| | |
|---|---|
| Reporting on sex and gender | No research involving human participants has been performed. |
| Reporting on race, ethnicity, or other socially relevant groupings | No research involving human participants has been performed. |
| Population characteristics | No research involving human participants has been performed. |
| Recruitment | No research involving human participants has been performed. |
| Ethics oversight | No research involving human participants has been performed. |

Note that full information on the approval of the study protocol must also be provided in the manuscript.

# Field-specific reporting

Please select the one below that is the best fit for your research. If you are not sure, read the appropriate sections before making your selection.

☒ Life sciences ☐ Behavioural & social sciences ☐ Ecological, evolutionary & environmental sciences

For a reference copy of the document with all sections, see nature.com/documents/nr-reporting-summary-flat.pdf

# Life sciences study design

All studies must disclose on these points even when the disclosure is negative.

| | |
|---|---|
| Sample size | Sample size calculations were not performed. Selected sample sizes were designed to ensure clear and reliable interpretation of the results. Based on previous experience in terms of variability, at least two independent replicates were carried out for all functional assays, as is standard for such experiments for other studies in this field (e.g. refs 9 & 10). |
| Data exclusions | No data were excluded. |
| Replication | All biochemical experiments were performed (at least) in technical duplicates along with appropriate controls, with successful replication of results in all cases. |
| Randomization | No grouped samples. |
| Blinding | No grouped samples. |

# Reporting for specific materials, systems and methods

We require information from authors about some types of materials, experimental systems and methods used in many studies. Here, indicate whether each material, system or method listed is relevant to your study. If you are not sure if a list item applies to your research, read the appropriate section before selecting a response.

## Materials & experimental systems

| n/a | Involved in the study |
|---|---|
| ☐ | ☒ Antibodies |
| ☐ | ☒ Eukaryotic cell lines |
| ☒ | ☐ Palaeontology and archaeology |
| ☒ | ☐ Animals and other organisms |
| ☒ | ☐ Clinical data |
| ☒ | ☐ Dual use research of concern |
| ☒ | ☐ Plants |

## Methods

| n/a | Involved in the study |
|---|---|
| ☒ | ☐ ChIP-seq |
| ☒ | ☐ Flow cytometry |
| ☒ | ☐ MRI-based neuroimaging |

## Antibodies

| | |
|---|---|
| Antibodies used | Anti-Ubiquitin (P4D1) mouse mAb (HRP conjugate), Cell Signaling Technology, Cat. No. 14049S, Lot 3; 1:5000 dilution |
| Validation | https://www.cellsignal.com/products/antibody-conjugates/ubiquitin-p4d1-mouse-mab-hrp-conjugate/14049 |

## Eukaryotic cell lines

Policy information about cell lines and Sex and Gender in Research

| | |
|---|---|
| Cell line source(s) | HEK293S GnTI- (identifier: CRL-3022) were obtained from ATCC. High five cells (BTI-TN-5B1-4) were obtained from ThermoFisher Scentific (catalogue number:B85502). Gibco Sf9 cells were obtained from ThermoFisher Scentific (catalogue number:11496016). |
| Authentication | Cell lines were not authenticated. |
| Mycoplasma contamination | Cell lines were periodically tested for mycoplasm contamination with no contamination detected. |
| Commonly misidentified lines (See ICLAC register) | No commonly misidentified cell lines were used in this study. |

## Plants

| | |
|---|---|
| Seed stocks | *Report on the source of all seed stocks or other plant material used. If applicable, state the seed stock centre and catalogue number. If plant specimens were collected from the field, describe the collection location, date and sampling procedures.* |
| Novel plant genotypes | *Describe the methods by which all novel plant genotypes were produced. This includes those generated by transgenic approaches, gene editing, chemical/radiation-based mutagenesis and hybridization. For transgenic lines, describe the transformation method, the number of independent lines analyzed and the generation upon which experiments were performed. For gene-edited lines, describe the editor used, the endogenous sequence targeted for editing, the targeting guide RNA sequence (if applicable) and how the editor was applied.* |
| Authentication | *Describe any authentication procedures for each seed stock used or novel genotype generated. Describe any experiments used to assess the effect of a mutation and, where applicable, how potential secondary effects (e.g. second site T-DNA insertions, mosiacism, off-target gene editing) were examined.* |

