## [Peer Review File · Nature Structural & Molecular Biology]

TRIP12 structures reveal HECT E3 formation of K29 linkages and branched ubiquitin chains

Corresponding Author: Professor Brenda Schulman

Version 0:

Decision Letter:

23rd Oct 2024

Dear Professor Schulman,

Thank you again for submitting your manuscript "TRIP12 ubiquitin chain-branching reveals consensus HECT E3 mechanism of polyubiquitylation". I apologise for the delay in responding, which resulted from the difficulty in timely obtaining suitable referee reports. Nevertheless, we now have comments (below) from the 3 reviewers who evaluated your paper. In light of those reports, we remain interested in your study and would like to see your response to the comments of the referees, in the form of a revised manuscript.

You will see that though the reviewers are generally appreciative of the advance imparted by the study, they nevertheless raise both technical and conceptual concerns that must be addressed in a revised manuscript. More specifically, the experts request numerous technical clarifications, additional analyses and controls, and improved presentation (e.g. Reviewer #1 on full-map quality). They also raise certain interesting functional questions (e.g. Reviewer #2 point 2) and ask for further discussion or caveating certain statements. We editorially agree that providing further experimental evidence and mechanistic granularity, as well as tempering certain conclusions on generalisability, would further boost the value of the manuscript. Please be sure to address/respond to all concerns of the referees in full in a point-by-point response and highlight all changes in the revised manuscript text file. If you have comments that are intended for editors only, please include those in a separate cover letter.

We expect to see your revised manuscript within 3 months. If you cannot send it within this time, please contact us to discuss an extension; we would still consider your revision, provided that no similar work has been accepted for publication at NSMB or published elsewhere.

Reporting Summary:

When submitting the revised version of your manuscript, please pay close attention to our <https://www.nature.com/nature-portfolio/editorial-policies/image-integrity> Digital Image Integrity Guidelines and to the following points below:

If there are additional or modified structures presented in the final revision, please submit the corresponding PDB validation reports, as well as the corresponding maps and models.

Data availability: this journal strongly supports public availability of data. All data used in accepted papers should be available via a public data repository, or alternatively, as Supplementary Information. If data can only be shared on request, please explain why in your Data Availability Statement, and also in the correspondence with your editor. Please note that for some data types, deposition in a public repository is mandatory - more information on our data deposition policies and available repositories can be found below:

<https://www.nature.com/nature-research/editorial-policies/reporting-standards#availability-of-data>

We require deposition of coordinates (and, in the case of crystal structures, structure factors) into the Protein Data Bank with the designation of immediate release upon publication (HPUB). Electron microscopy-derived density maps and coordinate data must be deposited in EMDb and released upon publication. Deposition and immediate release of NMR chemical shift assignments are highly encouraged. Deposition of deep sequencing and microarray data is mandatory, and the datasets must be released prior to or upon publication. To avoid delays in publication, dataset accession numbers must be supplied with the final accepted manuscript and appropriate release dates must be indicated at the galley proof stage.

Link Redacted

Sincerely,

Dimitris Typas
Senior Editor
Nature Structural & Molecular Biology
ORCID: 0000-0002-8737-1319

Reviewers' Comments:

Reviewer #1 (Remarks to the Author):

This study by Maiwald et al. provides significant structural and mechanistic insights into the process of ubiquitin chain branching by the HECT E3 ligase TRIP12. Using a combination of biochemistry, chemical biology, and cryo-EM, the authors elucidate the catalytic architecture responsible for forming K29-linked and K29/K48-branched ubiquitin chains. The work not only reveals the specific mechanism of TRIP12 but also suggests a potential consensus mechanism for linkage-specific polyubiquitylation by HECT E3 ligases in general.

The study is generally of high quality; however, some of the claims appear to be overstated. The high quality of the two delta N maps and model is particularly impressive. I recommend this study for publication in NSMB, but some improvements,

mainly in the presentation of results, are suggested.

Main concerns:

(1) The text related to Fig2c states "Importantly, mutations at all the testable of these interfaces impaired TRIP12-mediated branched Ub chain formation." These residues may also mediate K29 chain extension. It is unclear whether the introduced mutations specifically impair branched chain formation or would impair ubiquitination in general. Showing ubiquitination reactions focused on homotypic chains would support this statement.

(2) The Full-length map suffers from severe preferred orientation and is difficult to interpret. The resolution estimate appears inaccurate, and the text presentation of the next steps is confusing. I suggest to openly explain that this map is of poor quality and that interpretation beyond the overall shape is not possible and taking this as rational to make the truncated maps. The full-length map could be overlaid with the truncated maps to show that the overall shape is similar; otherwise, the value of this low-quality map is questionable.

(3) The abstract statement "The data define a consensus mechanism for linkage-specific polyubiquitylation by HECT E3s" seems overly strong. While the findings are important, they may not be applicable to all HECTs, given the known distinct structures, conformations, and potential auxiliary domains of different HECTs. It is suggested to rephrase this, as only two HECTs are discussed.

(4) On page 2, the statement about K29/K48 branched chains serving as superior signals for proteasomal degradation lacks appropriate citations. The statement is true for K11/K48. The cited reviews also don't support the claim and on a quick search I could not find evidence for the claim. Please add appropriate primary research citations for this claim.

(5) The explanation on page 3 regarding geometric constraints could be rephrased for clarity, particularly in relation to which distances are discussed and how they relate to the active site.

(6) The statement on page 4, "The arrangement of TRIP12's two Ub binding sites is compatible with a K48-linked chain but not other linkages," should be qualified to acknowledge that this applies to the particular conformation observed. Without data on apo-TRIP12, it's premature to rule out other chain topologies.

(7) In Extended Figures 3 and 4, no ab initio volume is marked as input for the processing pipelines. Please clarify or amend the processing pipeline to include all steps and specify the initial volume used.

(8) Map-model correlation should be included in the Table.

Minor concerns:

(9) The overall writing could be more engaging and simplified. Many sentences are long and dense, which could be significantly improved for readability.

(10) The use of 'substrates' to refer to ubiquitin combinations and variants may be confusing for general readers who typically consider the target protein as the substrate.

(11) It would be helpful to put all maps and models in the same coordinate system and orientation to allow readers to easily overlap them and see the differences.

Reviewer #2 (Remarks to the Author):

In this manuscript, Schulman and colleagues report the mechanism of ubiquitin chain branching by revealing the cryo-EM structure of TRIP12 in complex with Ub. They first showed that TRIP12 inserts K29-linkages into the proximal ubiquitin moiety of K48-linked di-Ub and that geometry of targeted Lys with TRIP12 is critical for the reaction. Next, using these information, they conducted sophisticated strategies to show cryo-EM structure of TRIP12 and ubiquitins. The obtained structure revealed interactions between HECT elements and donor/acceptor ubiquitins, how they are properly oriented to accomplish specific assembly of K29-linked chains. They also validated the importance of these interactions by mutational analyses.

Overall, this is a very interesting study with tremendous of implications for the mechanism of specific chain assembly by HECT ligases. They not only revealed the mechanism of branched ubiquitin chain formation but also visualized how HECT ligases generate specific ubiquitin chain linkages -e.g., accommodation of the acceptor ubiquitin (either mono- or di-ubiquitin), glueing of the acceptor and donor ubiquitins and the HECT C-lobe, and the role of the extreme C-terminal region of the E3.

I recommend the authors add some discussions or additional data regarding the following points for the broader readership from biology fields.

1. The cryo-EM structure of UFD4 in complex with ubiquitins has been reported. Some discussion will be informative

regarding the mechanistic differences (if any) between TRIP12 and UFD4.

2. It is reported that TRIP12 can assemble homotypic K29-linked ubiquitin chains in vitro. As the presented data show that K48-linked diUb serves as a better substrate than mono-Ub, I am wondering if TRIP12 usually adds multiple branched mono-Ubs from a K48-linked chain rather than it elongates a K29-linked chain from a K48-linked chain?

3. Related to this, Extended Fig. 1a-b shows that K48-linked diUb is superior to other linkages or mono-Ub as an acceptor, but how about the efficiency of other diUbs? Can the other diUbs not be used as acceptors due to steric hinderance, or they are as efficient as monoUb?

4. It has been enigmatic whether the extreme C-terminal amino acid of HECT E3s determines the linkage specificity (in this case, serine is conserved among TRIP12, UFD4, and UBE3C). Can the authors provide some insight regarding whether/how 'being serine' rather than the other amino acids is important for catalysis?

5. What insights does the mechanism of TRIP12 demonstrated in this study provide into the biological roles of TRIP12? As my understanding, TRIP12 should not act alone but in concert with other E3s assembling K48-linkages in cells.

6. Pluska et al (PMID: 33576509) reported the mechanism how UBE2K adds a branched K48-linkage from a K63-linkage. This paper should be cited.

Reviewer #3 (Remarks to the Author):

The manuscript by Maiwald and colleagues reports the structure and mechanism of the K29/K48 branching HECT E3 ubiquitin ligase TRIP12. While mechanisms of how HECT E3 ubiquitin ligases receive ubiquitin and transfer it onto substrates have been revealed, mechanisms of how branched ubiquitin chains are formed remain unknown. This is particularly relevant since branched (K29/K48) ubiquitin chains have emerged as enhanced signals for proteasomal degradation, although it is unclear how specific branched chains can be assembled. Given that 10% of all poly-ubiquitin chains contain branches, understanding the mechanisms on specific branched chain assembly is essential. Here Maiwald and colleagues present careful biochemical data and highly compelling structural data that reveals how TRIP12 assembles K29/K48 branches.

Overall this reviewer felt that the biochemical and structural data are solid and provide the first detailed molecular explanation for the formation of K29/K48 branches that may be applicable to other HECT E3 ligases.

Several minor comments are listed below that require additional clarification around the underlying mechanism proposed.

The biochemical data presented in Extended Figures 1a and b demonstrate only K48 diUb can be extended by TRIP12 and this requires distal Ub binding to the HECT domain. However, TRIP12 can assemble K29 diUb (albeit at longer timescales), thus mono Ub must clearly be able to bind and be positioned with K29 on the acceptor surface. Would this not imply that at longer time points linkages other than K48 can be extended with K29 chains?

The text mentions 'tight geometric restraints' required for branch formation, but donor Ub is less well resolved in the maps. Further analysis could determine if this is due to occupancy of donor Ub, flexibility (or tight geometric restraints only apply to the C-terminus of the donor Ub?). Further processing may help discern if low resolution for donor Ub is due to occupancy or flexibility: e.g. cryoSPARC/RELION 3D classification, occuPy, cryoSPARC 3D variability analysis.

The argument for improving resolution through removing conformation heterogeneity is somewhat confusing and misleading. The authors report an overall 3.2 Å structure of full-length TRIP12 bound to a stable K48/K29 mimetic triUb, yet report a 3.7 Å structure of a truncated TRIP12 lacking 477 flexible residues. While these are overall resolutions, and the local resolution of full-length TRIP12 is worse around the regions of interest, the authors should rephrase the second paragraph on page 4 to state that the local resolution is much improved from a truncated TRIP12 and this is likely due to better particle distribution on the grids due to removal of the N-terminal disordered regions that do not impact upon TRIP12 K29 branching activity. The methods section on page 21 mentions anisotropic maps due to preferred particle orientation and this is clear from the particle distribution plot in Extended Figure 2c, but this should be made clearer in the main text.

A recent preprint describing the mechanism of action of the yeast orthologue of TRIP12, Ufd4 has been recently reported (biorxiv: 2023.05.23.542033) and is acknowledged by the authors. However, the final paragraph 'Thus, although some HECT E3s may utilize alternative modes of binding to the donor and acceptor Ubs....' is not relevant to this reference (ref 38) as this preprint does not compare to other HECT E3s.

Examples of model fit to density are missing, especially for areas of interest such as the active site and Ub branch point and should be included in the extended figures.

Finally, "forging" is somewhat overused in the manuscript to describe the generation of K29 linkages and K29/K48 branches. The authors should occasionally use a different verb.

Version 1:

Decision Letter:

Our ref: NSMB-BC49696A

11th Feb 2025

Dear Professor Schulman,

Thank you for submitting your revised manuscript "TRIP12 ubiquitin chain-branching reveals conserved HECT E3 mechanism of polyubiquitylation" (NSMB-BC49696A). I apologise for the slight delay in returning our decision. Nevertheless, the manuscripts has now been seen by the original referees and their comments are below. The reviewers find that the paper has improved in revision, and therefore we are happy to accept it in principle in Nature Structural & Molecular Biology, pending minor revisions to satisfy the referees' final requests with respect to the perceived overstatement about the generalisability of the mechanism and to comply with our editorial and formatting guidelines.

We are now performing detailed checks on your paper and will send you a checklist detailing our editorial and formatting requirements in about two weeks. Please do not upload the final materials and make any revisions until you receive this additional information from us.

To facilitate our work at this stage, it is important that we have a copy of the main text as a word file. If you could please send along a word version of this file as soon as possible, we would greatly appreciate it; please make sure to copy the NSMB account (cc'ed above).

Sincerely,

Dimitris Typas
Senior Editor
Nature Structural & Molecular Biology
ORCID: 0000-0002-8737-1319

Reviewer #1 (Remarks to the Author):

I greatly appreciate the authors' thorough response to my previous comments and their substantial improvements to the manuscript. The work is of high quality and merits publication. However, I would like to bring attention to two remaining points that would benefit from further refinement:

Regarding the title's scope: While the study provides valuable insights into chain branching mechanisms, for TRIP12, the current evidence base may not fully support claims of a broader consensus mechanism. I recommend adjusting the language to more precisely reflect the scope of the findings also in the title, which would maintain scientific rigor without diminishing the manuscript's significant contributions to our understanding of chain branching.

Concerning the discussion of K29/K48 branches as enhanced proteasomal substrates: Now multiple citations are given. I note that many of these references hardly mention the cited statement and some are reviews without direct experimental evidence. For such a specific mechanistic claim, I would suggest citing primary sources that include direct measurements of this phenomenon. This would strengthen the manuscript's scientific foundation and again show more scientific rigor. To be frank, the statement isn't necessary for the manuscript so it even could be taken out.

With these minor revisions, I believe the manuscript will be ready for publication. The overall quality of the experimental work remains impressive.

Reviewer #2 (Remarks to the Author):

In the revised manuscript, Dr. Schulman and colleagues adequately addressed my previous concerns by adding new experiments and discussions. The new data characterizing the enzymatic properties of TRIP12 are impressive. Also, new discussion regarding the interplay of TRIP12 with E3s generating K48-linked chains is informative to the readers. I highly recommend publication of the manuscript in Nature SMB.

Reviewer #3 (Remarks to the Author):

The authors have thoroughly responded to my feedback. Adding extra biochemical validation through the DUB assay and enhancing the structural presentation significantly bolster the conclusions drawn by the authors.

Version 2:

Decision Letter:

10th Apr 2025

Dear Professor Schulman,

We are now happy to accept your revised paper "TRIP12 structures reveal HECT E3 formation of K29 linkages and branched ubiquitin chains" for publication as an Article in Nature Structural & Molecular Biology.

Your paper will be published online soon after we receive proof corrections and will appear in print in the next available issue. You can find out your date of online publication by contacting the production team shortly after sending your proof corrections.

Sincerely,

Dimitris Typas
Senior Editor
Nature Structural & Molecular Biology
ORCID: 0000-0002-8737-1319

Responses to Reviewer comments are in blue.

Reviewers' Comments:

Reviewer #1 (Remarks to the Author):

This study by Maiwald et al. provides significant structural and mechanistic insights into the process of ubiquitin chain branching by the HECT E3 ligase TRIP12. Using a combination of biochemistry, chemical biology, and cryo-EM, the authors elucidate the catalytic architecture responsible for forming K29-linked and K29/K48-branched ubiquitin chains. The work not only reveals the specific mechanism of TRIP12 but also suggests a potential consensus mechanism for linkage-specific polyubiquitylation by HECT E3 ligases in general.

The study is generally of high quality; however, some of the claims appear to be overstated. The high quality of the two delta N maps and model is particularly impressive. I recommend this study for publication in NSMB, but some improvements, mainly in the presentation of results, are suggested.

We thank the reviewer for such kind and helpful comments and for enthusiasm for our study!

Main concerns:

(1) The text related to Fig2c states "Importantly, mutations at all the testable of these interfaces impaired TRIP12-mediated branched Ub chain formation." These residues may also mediate K29 chain extension. It is unclear whether the introduced mutations specifically impair branched chain formation or would impair ubiquitination in general. Showing ubiquitination reactions focused on homotypic chains would support this statement.

In response to the Reviewer, we performed assays with a high concentration of mono-Ub acceptor to assess effects of mutations on forming K29-linked di-Ub. As predicted by the Reviewer, and in agreement with our structural data, all mutants in the catalytic interfaces are defective in K29 chain formation. Only mutation of the binding site for the K48-linked distal Ub had no impact on K29 chain extension. We extended the text to provide greater insight into the mechanism, and show the new experiments in Extended Data Fig. 8 of the revised manuscript.

(2) The Full-length map suffers from severe preferred orientation and is difficult to interpret. The resolution estimate appears inaccurate, and the text presentation of the next steps is confusing. I suggest to openly explain that this map is of poor quality and that interpretation beyond the overall shape is not possible and taking this as rationale to make the truncated maps. The full-length map could be overlaid with the truncated maps to show that the overall shape is similar; otherwise, the value of this low-quality map is questionable.

We appreciate this suggestion. We meant to make this point in our original manuscript. Retrospectively, in preparing a Brief Communication, we inadvertently omitted some details raised by reviewers. We now provide clarification in two ways. In response to this Reviewer, and related comments from Reviewer #3, we have substantially revised the description of the map with full-length TRIP12. We extended the text, and describe that the resolution is limited due to challenges of preferred orientation. We also now show rotation around the maps in Supplementary Video 1. The video clarifies the limited quality of the full-length TRIP12 map, but that it fits well with the structure derived from the high-quality data obtained with TRIP12ΔN.

(3) The abstract statement "The data define a consensus mechanism for linkage-specific polyubiquitylation by HECT E3s" seems overly strong. While the findings are important, they may not be applicable to all HECTs, given the known distinct structures, conformations, and potential

auxiliary domains of different HECTs. It is suggested to rephrase this, as only two HECTs are discussed.

We thank the reviewer for this comment. We have rephrased the title, and expanded our Abstract, Introduction and Discussion to clarify that the conserved mechanism is derived from comparing two human HECT E3s, TRIP12 and UBR5. In the Discussion, we point out that the interface between the C-lobe and the donor Ub that we observe in the catalytic assembly with TRIP12 matches that in all prior structures with a HECT catalytic Cys covalently-linked to ubiquitin (and a prior crystal structure of an E2~ubiquitin-HECT domain complex) with the exception of the Ufd4 complex. This is depicted in Extended Data Fig. 10 of the revised manuscript. We also describe prior mutational analyses of acceptor ubiquitin recognition, which are consistent with the consensus mechanism.

Also, based on this suggestion and comments from the other Reviewers, we added discussion of the different donor and acceptor arrangement with Ufd4. We note that the structure with yeast Ufd4 was obtained with a mixed species complex (with human ubiquitin). The three residues differing in yeast Ub correspond to human acceptor Ub residues at the interface with TRIP12's C-lobe.

(4) On page 2, the statement about K29/K48 branched chains serving as superior signals for proteasomal degradation lacks appropriate citations. The statement is true for K11/K48. The cited reviews also don't support the claim and on a quick search I could not find evidence for the claim. Please add appropriate primary research citations for this claim.

We have rewritten the Introduction and Discussion for clarity. We now focus on the demonstrated roles of TRIP12 and K29/K48 branched chains, and have added the citations.

(5) The explanation on page 3 regarding geometric constraints could be rephrased for clarity, particularly in relation to which distances are discussed and how they relate to the active site.

The revised manuscript clarifies that the experiments with semi-synthetic Ubs test the number of methylene groups between the alpha carbon and acceptor amino group that is the site of modification. We also clarify the specific distances maintained between the native transition state and our chemically-linked complexes.

(6) The statement on page 4, "The arrangement of TRIP12's two Ub binding sites is compatible with a K48-linked chain but not other linkages," should be qualified to acknowledge that this applies to the particular conformation observed. Without data on apo-TRIP12, it's premature to rule out other chain topologies.

We apologize for lack of clarity in our original text. The revision now clarifies as follows: "The tandem Ub-binding domain organization is observed even in the complex representing modification of a mono-Ub (Fig. 2d, Supplementary Video 1). The positioning of the adjacent, vacant site ready to capture the distal Ub in a K48-linked chain even in its absence rationalizes the striking preference for branching off of chains linked through K48, but not other lysines."

(7) In Extended Figures 3 and 4, no ab initio volume is marked as input for the processing pipelines. Please clarify or amend the processing pipeline to include all steps and specify the initial volume used.

We apologize for this oversight in our initial manuscript and thank the reviewer for this important suggestion. We now clarify in the Methods that we obtained the ab initio volumes in our screening datasets, which were then used as inputs for heterogeneous refinements in processing the higher resolution data. We added an Extended Data Fig. 3 showing the processing scheme for the screening datasets, and now include all steps and specify the reference volume used in the processing schemes in Extended Data Figs. 4 and 5.

(8) Map-model correlation should be included in the Table.

We apologize for this oversight in our initial manuscript and thank the reviewer for this important suggestion. We added this information to the Table.

Minor concerns:

(9) The overall writing could be more engaging and simplified. Many sentences are long and dense, which could be significantly improved for readability.

In preparing the revised manuscript, we shortened many sentences. We also extended the Introduction and Discussion to better reflect the exciting nature of the study.

(10) The use of 'substrates' to refer to ubiquitin combinations and variants may be confusing for general readers who typically consider the target protein as the substrate.

In the revised manuscript, we have largely replaced the term 'substrates' and now refer to the acceptor moiety.

(11) It would be helpful to put all maps and models in the same coordinate system and orientation to allow readers to easily overlap them and see the differences.

We provide reviewers with the maps and models aligned in a ChimeraX session at the following link: <https://datashare.biochem.mpg.de/s/Vm2fiFZfkxWkFLt>

Maps and models in the same coordinate system and orientation will be available from the EMDB and RCSB upon publication.

Reviewer #2 (Remarks to the Author):

In this manuscript, Schulman and colleagues report the mechanism of ubiquitin chain branching by revealing the cryo-EM structure of TRIP12 in complex with Ub. They first showed that TRIP12 inserts K29-linkages into the proximal ubiquitin moiety of K48-linked di-Ub and that geometry of targeted Lys with TRIP12 is critical for the reaction. Next, using these information, they conducted sophisticated strategies to show cryo-EM structure of TRIP12 and ubiquitins. The obtained structure revealed interactions between HECT elements and donor/acceptor ubiquitins, how they are properly oriented to accomplish specific assembly of K29-linked chains. They also validated the importance of these interactions by mutational analyses.

Overall, this is a very interesting study with tremendous of implications for the mechanism of specific chain assembly by HECT ligases. They not only revealed the mechanism of branched ubiquitin chain formation but also visualized how HECT ligases generate specific ubiquitin chain linkages -e.g., accommodation of the acceptor ubiquitin (either mono- or di-ubiquitin), glueing of the acceptor and donor ubiquitins and the HECT C-lobe, and the role of the extreme C-terminal region of the E3.

We thank the Reviewer for such kind and helpful comments and for enthusiasm for our study!

I recommend the authors add some discussions or additional data regarding the following points for the broader readership from biology fields.

1. The cryo-EM structure of UFD4 in complex with ubiquitins has been reported. Some discussion will be informative regarding the mechanistic differences (if any) between TRIP12 and UFD4.

In response to this suggestion and comments from both of the other reviewers, we more extensively compare our data for TRIP12 with existing data for other HECT E3s, including Ufd4.

Our revised manuscript now shows that the interface between the C-lobe and the donor Ub that we observe in the catalytic assembly with TRIP12 matches that in all prior structures with a HECT catalytic Cys covalently-linked to ubiquitin (and a prior crystal structure of an E2~ubiquitin-HECT domain complex) with the exception of the Ufd4 complex. We added discussion of the different donor and acceptor arrangement with Ufd4. We note that the structure with yeast Ufd4 was obtained with a mixed species complex (with human ubiquitin). The three residues differing in yeast Ub correspond to human acceptor Ub residues at the interface with TRIP12's C-lobe.

2. It is reported that TRIP12 can assemble homotypic K29-linked ubiquitin chains in vitro. As the presented data show that K48-linked diUb serves as a better substrate than mono-Ub, I am wondering if TRIP12 usually adds multiple branched mono-Ubs from a K48-linked chain rather than it elongates a K29-linked chain from a K48-linked chain?

To address this and a related question from Reviewer #3, we performed new experiments in Extended data Fig. 9 of the revision. The new text describing these experiments is pasted here:

"The structure suggests TRIP12 could engage - and thus target - di-Ubs along a K48-linked chain. We tested this using pulse-chase assays examining TRIP12 modification of longer chains. K29-linked chains (largely tetra-Ub, but with some tri-Ub) and K48-linked chains (primarily tetra-Ub and a minor amount of penta-Ub) were biochemically prepared for testing as TRIP12 substrates. Ubiquitylation was initiated with E2~*Ub(K0) either equimolar with or in 4-fold excess of acceptors. Mono-Ub was converted to di-Ub, but modification of the K29-linked chain was not readily detected. Strikingly, however, TRIP12 robustly added multiple mono-Ubs to the K48-linked chains (**Extended Data Fig. 9a**). We also compared TRIP12 activity with E2~Ub complexes harboring either untagged K29R Ub or WT Ub as the donor, as the latter in principle permits its subsequent use as an acceptor during polyubiquitylation (**Extended Data Fig. 9b**). For these assays, immunoblotting with anti-Ub antibodies detects Ubs from both the donor and the acceptor moieties. The products generated with the K29R-linked donor Ub largely resembled those in assays with *Ub(K0), while additional bands and variations in their relative intensities were observed in assays with WT donor Ub. Thus, to determine the nature of these polyubiquitin chains, we tested their cleavage by linkage-specific deubiquitylating (DUB) enzymes (**Extended Data Fig. 9c**). The main products were confirmed as mono-Ubs linked to K29 of acceptors in the K48-linked chain because treatment with OTUB1* (a K48-linkage specific DUB¹⁶) collapsed the majority of products to di-Ub. Susceptibility of the OTUB1*-generated di-Ub products to deconjugation by TRABID (a K29-linkage specific DUB^{16,17}) confirmed the K29-linkages. Taken together, the data support the conclusion that TRIP12 preferentially branches K29 linkages from K48-linked di-Ubs, including within in a polyUb chain."

3. Related to this, Extended Fig. 1a-b shows that K48-linked diUb is superior to other linkages or mono-Ub as an acceptor, but how about the efficiency of other diUbs? Can the other diUbs not be used as acceptors due to steric hinderance, or they are as efficient as monoUb?

To address this, we now assay a suite of potential acceptors at two different concentrations (Fig. 1a of the revised manuscript). The data show TRIP12 preferentially targets K48-linked chains compared to di-Ubs with any other linkage or mono-Ub. In experiments performed with the higher acceptor concentration (1 micromolar), some activity was observed towards mono-Ub and di-Ubs with K6, K11, K63 linkages, but not others (M1, K27, K29, and K33). The revised manuscript shows the structural explanation, in Extended Data Fig. 7. No other linkage besides K48 is compatible with occupation of the tandem Ub binding site in the ARM domain (Extended Data Fig. 7c). Meanwhile, Extended Data Fig. 7d of the revised manuscript shows that M1, K33, and (obviously) K29 abut the active site and thus their linkage would prevent utilization as an acceptor, while K27 is buried.

4. It has been enigmatic whether the extreme C-terminal amino acid of HECT E3s determines the

linkage specificity (in this case, serine is conserved among TRIP12, UFD4, and UBE3C). Can the authors provide some insight regarding whether/how 'being serine' rather than the other amino acids is important for catalysis?

TRIP12's C-terminal Ser fits snugly in the interface between TRIP12's N- and C-lobes, and the acceptor Ub, and aligns the donor Ub's covalent bond with TRIP12. To address the question regarding the importance of its identity, we generated versions of TRIP12 with the C-terminal Ser replaced by the other side-chains found at HECT E3 C-termini (A, D, E, H, I, L, N, V and Y). All of these are defective in ubiquitylation, not only of K48-linked di-Ub, but also of mono-Ub. The global defects are consistent with the tightly packed active site, and the previously proposed structural and catalytic roles of a HECT E3 C-terminal residue. These new data are shown in Fig. 4a and Extended Data Figure 8 of the revised manuscript.

5. What insights does the mechanism of TRIP12 demonstrated in this study provide into the biological roles of TRIP12? As my understanding, TRIP12 should not act alone but in concert with other E3s assembling K48-linkages in cells.

We thank the reviewer for this excellent question. We address this in the revised manuscript in the expanded Introduction and Discussion sections. The revised manuscript describes prior studies linking TRIP12 to cellular pathways also involving other E3 ligases that assemble K48-linkages. We also point out that TRIP12's top co-dependencies in DEPMAP include UBR5 and UBE2R2, which are E3 and E2 enzymes that selectively produce K48-linked chains and that have been biologically linked to TRIP12.

6. Pluska et al (PMID: 33576509) reported the mechanism how UBE2K adds a branched K48-linkage from a K63-linkage. This paper should be cited.

We are very happy to include this citation, which is now in the expanded Introduction section of our revised manuscript.

Reviewer #3 (Remarks to the Author):

The manuscript by Maiwald and colleagues reports the structure and mechanism of the K29/K48 branching HECT E3 ubiquitin ligase TRIP12. While mechanisms of how HECT E3 ubiquitin ligases receive ubiquitin and transfer it onto substrates have been revealed, mechanisms of how branched ubiquitin chains are formed remain unknown. This is particularly relevant since branched (K29/K48) ubiquitin chains have emerged as enhanced signals for proteasomal degradation, although it is unclear how specific branched chains can be assembled. Given that 10% of all poly-ubiquitin chains contain branches, understanding the mechanisms on specific branched chain assembly is essential. Here Maiwald and colleagues present careful biochemical data and highly compelling structural data that reveals how TRIP12 assembles K29/K48 branches.

Overall this reviewer felt that the biochemical and structural data are solid and provide the first detailed molecular explanation for the formation of K29/K48 branches that may be applicable to other HECT E3 ligases.

We thank the reviewer for such kind and helpful comments and for enthusiasm for our study!

Several minor comments are listed below that require additional clarification around the underlying mechanism proposed.

The biochemical data presented in Extended Figures 1a and b demonstrate only K48 diUb can be extended by TRIP12 and this requires distal Ub binding to the HECT domain. However, TRIP12 can assemble K29 diUb (albeit at longer timescales), thus mono Ub must clearly be able to bind and be

positioned with K29 on the acceptor surface. Would this not imply that at longer time points linkages other than K48 can be extended with K29 chains?

To address this and related questions from Reviewer #2, we performed several experiments that further define TRIP12's preferred acceptors. We now assay the full suite of di-Ubs as acceptors, at two different concentrations (Fig. 1a of the revised manuscript). The data show TRIP12 preferentially targets K48-linked chains compared to di-Ubs with any other linkage or mono-Ub. In experiments performed with the higher acceptor concentration (1 micromolar), some activity was observed towards mono-Ub and di-Ubs with K6, K11, K63 linkages, but not others (M1, K27, K29, and K33). The revised manuscript shows the structural explanation, in Extended Data Fig. 7. No other linkage besides K48 is compatible with occupation of the tandem Ub binding site in the ARM domain (Extended Data Fig. 7c). Meanwhile, Extended Data Fig. 7d of the revised manuscript shows that M1, K33, and (obviously) K29 about the active site and thus their linkage would prevent utilization as an acceptor, while K27 is buried.

We also performed new experiments comparing TRIP12 activity towards longer chains, shown in Extended Data Fig. 9 in the revision. The new text describing these experiments is pasted here:

"The structure suggests TRIP12 could engage - and thus target - di-Ubs along a K48-linked chain. We tested this using pulse-chase assays examining TRIP12 modification of longer chains. K29-linked chains (largely tetra-Ub, but with some tri-Ub) and K48-linked chains (primarily tetra-Ub and a minor amount of penta-Ub) were biochemically prepared for testing as TRIP12 substrates. Ubiquitylation was initiated with E2~*Ub(K0) either equimolar with or in 4-fold excess of acceptors. Mono-Ub was converted to di-Ub, but modification of the K29-linked chain was not readily detected. Strikingly, however, TRIP12 robustly added multiple mono-Ubs to the K48-linked chains (**Extended Data Fig. 9a**). We also compared TRIP12 activity with E2~Ub complexes harboring either untagged K29R Ub or WT Ub as the donor, as the latter in principle permits its subsequent use as an acceptor during polyubiquitylation (**Extended Data Fig. 9b**). For these assays, immunoblotting with anti-Ub antibodies detects Ubs from both the donor and the acceptor moieties. The products generated with the K29R-linked donor Ub largely resembled those in assays with *Ub(K0), while additional bands and variations in their relative intensities were observed in assays with WT donor Ub. Thus, to determine the nature of these polyubiquitin chains, we tested their cleavage by linkage-specific deubiquitylating (DUB) enzymes (**Extended Data Fig. 9c**). The main products were confirmed as mono-Ubs linked to K29 of acceptors in the K48-linked chain because treatment with OTUB1* (a K48-linkage specific DUB¹⁶) collapsed the majority of products to di-Ub. Susceptibility of the OTUB1*-generated di-Ub products to deconjugation by TRABID (a K29-linkage specific DUB^{16,17}) confirmed the K29-linkages. Taken together, the data support the conclusion that TRIP12 preferentially branches K29 linkages from K48-linked di-Ubs, including within in a polyUb chain."

The text mentions 'tight geometric restraints' required for branch formation, but donor Ub is less well resolved in the maps. Further analysis could determine if this is due to occupancy of donor Ub, flexibility (or tight geometric restraints only apply to the C-terminus of the donor Ub?). Further processing may help discern if low resolution for donor Ub is due to occupancy or flexibility: e.g. cryoSPARC/RELION 3D classification, occuPy, cryoSPARC 3D variability analysis.

To address this, we now describe the 3D variability analysis using CryoSPARC in the Main Text and show the results as Supplementary Video 2. To highlight the variations in conformations, we fit the maps with the following units extracted from the final refined coordinate file: (1) the TRIP12 ARM domain, (2) the TRIP12 HEL-UBL domain and HECT domain N-lobe, (3) the distal Ub, and (4) the TRIP12 HECT domain C-lobe/C-terminus and acceptor and donor Ubs. As shown in the Video, the frames successively show increased visibility not only of the donor Ub, but also TRIP12's extreme C-terminal region (including the notoriously key residues for HECT E3-mediated polyubiquitylation, here the -4Phe, -2Leu, and C-terminal Ser). This transition is concomitant with progression of the C-lobe/acceptor Ub unit towards the N-lobe. As such, the donor Ub and TRIP12 C-terminus are maximally visible in the configuration where the key TRIP12 N-lobe, C-lobe, C-terminal tail, and

donor Ub and acceptor Ub elements required for polyubiquitylation are intertwined. We selected for this latter state through 3D classification, where it yielded the highest resolution map (Extended Data Figure 4). We focus the main text on this class because it best visualizes the key TRIP12 elements and Ub residues required for polyubiquitylation.

The argument for improving resolution through removing conformation heterogeneity is somewhat confusing and misleading. The authors report an overall 3.2 Å structure of full-length TRIP12 bound to a stable K48/K29 mimetic triUb, yet report a 3.7 Å structure of a truncated TRIP12 lacking 477 flexible residues. While these are overall resolutions, and the local resolution of full-length TRIP12 is worse around the regions of interest, the authors should rephrase the second paragraph on page 4 to state that the local resolution is much improved from a truncated TRIP12 and this is likely due to better particle distribution on the grids due to removal of the N-terminal disordered regions that do not impact upon TRIP12 K29 branching activity. The methods section on page 21 mentions anisotropic maps due to preferred particle orientation and this is clear from the particle distribution plot in Extended Figure 2c, but this should be made clearer in the main text.

We appreciate this suggestion. We meant to make this point in our original manuscript. Retrospectively, in preparing a Brief Communication, we inadvertently omitted some details raised by reviewers. We now provide clarification in two ways. In response to this Reviewer, and related comments from Reviewer #3, we have substantially revised the description of the map with full-length TRIP12. We extended the text, and describe that the resolution is limited due to challenges of preferred orientation. We also now show rotation around the maps in Supplementary Video 1. The video clarifies the limited quality of the full-length TRIP12 map, but that it fits well with the structure derived from the high-quality data obtained with TRIP12ΔN.

A recent preprint describing the mechanism of action of the yeast orthologue of TRIP12, Ufd4 has been recently reported (biorxiv: 2023.05.23.542033) and is acknowledged by the authors. However, the final paragraph 'Thus, although some HECT E3s may utilize alternative modes of binding to the donor and acceptor Ubs....' is not relevant to this reference (ref 38) as this preprint does not compare to other HECT E3s.

In response to this suggestion and comments from both of the other reviewers, we more extensively compare our data for TRIP12 with existing data for other HECT E3s. We revised the text to more precisely discuss Ufd4.

Examples of model fit to density are missing, especially for areas of interest such as the active site and Ub branch point and should be included in the extended figures.

We apologize for this omission from our original manuscript. We now include Extended Data Fig. 6, which provides several examples of model fit to the density around the active site for TRIP12ΔN generating the K29 linkage. Supplementary Video 1 also shows the overall fit of coordinates docked in the maps for full-length TRIP12 generating a K29/K48-linked branched chain, TRIP12ΔN producing a K29/K48-linked branched chain, and TRIP12ΔN synthesizing a K29 linkage to a mono-Ub.

Finally, "forging" is somewhat overused in the manuscript to describe the generation of K29 linkages and K29/K48 branches. The authors should occasionally use a different verb.

In the revision, we replaced the majority of "forging" (and "forge", etc.) usages with alternative terms.

Response to Reviewer comments are in blue.

Reviewer #1 (Remarks to the Author):

I greatly appreciate the authors' thorough response to my previous comments and their substantial improvements to the manuscript. The work is of high quality and merits publication. However, I would like to bring attention to two remaining points that would benefit from further refinement:

Regarding the title's scope: While the study provides valuable insights into chain branching mechanisms, for TRIP12, the current evidence base may not fully support claims of a broader consensus mechanism. I recommend adjusting the language to more precisely reflect the scope of the findings also in the title, which would maintain scientific rigor without diminishing the manuscript's significant contributions to our understanding of chain branching.

In the title, we removed the word “conserved”, which was the issue referred to by the Reviewer. The title now is: “TRIP12 structures reveal HECT E3 formation of K29 linkages and branched ubiquitin chains”.

Concerning the discussion of K29/K48 branches as enhanced proteasomal substrates: Now multiple citations are given. I note that many of these references hardly mention the cited statement and some are reviews without direct experimental evidence. For such a specific mechanistic claim, I would suggest citing primary sources that include direct measurements of this phenomenon. This would strengthen the manuscript's scientific foundation and again show more scientific rigor. To be frank, the statement isn't necessary for the manuscript so it even could be taken out.

We replaced the sentence with one specifically summarizing the 6 primary literature references that were cited. We moved the citations to the two review articles to the more general statement in the next sentence. In particular and as requested by the reviewer, we removed any potential implication of the mechanism by which K29/K48-linked branched chains impact the regulation. These two sentences now read:

Meanwhile, branched chains with both K29 and K48 linkages have been implicated in the regulation of diverse substrates, in biological processes ranging from responses to oxidative, lipid, and pH stresses to targeted protein degradation 20-25. Despite great biological importance^{26,27}, our understanding of how E3s generate K29-linked Ub chains and linkage-specific branched chains remains limited.

With these minor revisions, I believe the manuscript will be ready for publication. The overall quality of the experimental work remains impressive.

We thank the Reviewer for their kind and helpful comments and suggestions, and appreciation of our study!

Reviewer #2 (Remarks to the Author):

In the revised manuscript, Dr. Schulman and colleagues adequately addressed my previous concerns by adding new experiments and discussions. The new data characterizing the enzymatic properties of TRIP12 are impressive. Also, new discussion regarding the interplay of TRIP12 with E3s generating K48-linked chains is informative to the readers. I highly recommend publication of the manuscript in Nature SMB.

We thank the Reviewer for their kind and helpful comments and suggestions, and appreciation of our study!

Reviewer #3 (Remarks to the Author):

The authors have thoroughly responded to my feedback. Adding extra biochemical validation through the DUB assay and enhancing the structural presentation significantly bolster the conclusions drawn by the authors.

We thank the Reviewer for their kind and helpful comments and suggestions, and appreciation of our study!